# LEARNING TOWARDS THE LARGEST MARGINS

**Xiong Zhou**[1*]**, Xianming Liu**[1,2†]**, Deming Zhai**[1]**, Junjun Jiang**[1,2]**, Xin Gao**[3,2,4]**, Xiangyang Ji**[5]
[1]Harbin Institute of Technology  [2]Peng Cheng Laboratory  [3]King Abdullah University of Science and Technology
[4]Gaoling School of Artificial Intelligence, Renmin University of China  [5]Tsinghua University

## ABSTRACT

One of the main challenges for feature representation in deep learning-based classification is the design of appropriate loss functions that exhibit strong discriminative power. The classical softmax loss does not explicitly encourage discriminative learning of features. A popular direction of research is to incorporate margins in well-established losses in order to enforce extra intra-class compactness and inter-class separability, which, however, were developed through heuristic means, as opposed to rigorous mathematical principles. In this work, we attempt to address this limitation by formulating the principled optimization objective as *learning towards the largest margins*. Specifically, we firstly define the class margin as the measure of inter-class separability, and the sample margin as the measure of intra-class compactness. Accordingly, to encourage discriminative representation of features, the loss function should promote the largest possible margins for both classes and samples. Furthermore, we derive a generalized margin softmax loss to draw general conclusions for the existing margin-based losses. Not only does this principled framework offer new perspectives to understand and interpret existing margin-based losses, but it also provides new insights that can guide the design of new tools, including *sample margin regularization* and *largest margin softmax loss* for the class-balanced case, and *zero-centroid regularization* for the class-imbalanced case. Experimental results demonstrate the effectiveness of our strategy on a variety of tasks, including visual classification, imbalanced classification, person re-identification, and face verification.

## 1 INTRODUCTION

Recent years have witnessed the great success of deep neural networks (DNNs) in a variety of tasks, especially for visual classification (Simonyan & Zisserman, 2014; Szegedy et al., 2015; He et al., 2016; Howard et al., 2017; Zoph et al., 2018; Touvron et al., 2019; Brock et al., 2021; Dosovitskiy et al., 2021). The improvement in accuracy is attributed not only to the use of DNNs, but also to the elaborated losses encouraging well-separated features (Elsayed et al., 2018; Musgrave et al., 2020).

In general, the loss is expected to promote the learned features to have maximized intra-class compactness and inter-class separability simultaneously, so as to boost the feature discriminativeness. Softmax loss, which is the combination of a linear layer, a softmax function, and cross-entropy loss, is the most commonly-used ingredient in deep learning-based classification. However, the softmax loss only learns separable features that are not discriminative enough (Liu et al., 2017). To remedy the limitation of softmax loss, many variants have been proposed. Liu et al. (2016) proposed a generalized large-margin softmax loss, which incorporates a preset constant $m$ multiplying with the angle between samples and the classifier weight of the ground truth class, leading to potentially larger angular separability between learned features. SphereFace (Liu et al., 2017) further improved the performance of L-Softmax by normalizing the prototypes in the last inner-product layer. Subsequently, Wang et al. (2017) exhibited the usefulness of feature normalization when using feature vector dot products in the softmax function. Coincidentally, in the field of contrastive learning, Chen et al. (2020) also showed that normalization of outputs leads to superior representations. Due to its effectiveness, normalization on either features or prototypes or both becomes a standard procedure in margin-based losses, such as SphereFace (Liu et al., 2017), CosFace/AM-Softmax (Wang et al., 2018b;a) and ArcFace (Deng et al., 2019). However, there is no theoretical guarantee provided yet.

---

*This work was done as intern at Peng Cheng Laboratory.
†Correspondence to: Xianming Liu <csxm@hit.edu.cn>

Despite their effectiveness and popularity, the existing margin-based losses were developed through heuristic means, as opposed to rigorous mathematical principles, modeling and analysis. Although they offer geometric interpretations, which are helpful to understand the underlying intuition, the theoretical explanation and analysis that can guide the design and optimization is still vague. Some critical issues are unclear, *e.g.*, why is the normalization of features and prototypes necessary? How can the loss be further improved or adapted to new tasks? Therefore, it naturally raises a fundamental question: how to develop a principled mathematical framework for better understanding and design of margin-based loss functions? The goal of this work is to address these questions by formulating the objective as learning towards the largest margins and offering rigorously theoretical analysis as well as extensive empirical results to support this point.

To obtain an optimizable objective, firstly, we should define measures of intra-class compactness and inter-class separability. To this end, we propose to employ the class margin as the measure of inter-class separability, which is defined as the minimal pairwise angle distance between prototypes that reflects the angular margin of the two closest prototypes. Moreover, we define the sample margin following the classic approach in (Koltchinskii et al., 2002, Sec. 5), which denotes the similarity difference of a sample to the prototype of the class it belongs to and to the nearest prototype of other classes and thus measures the intra-class compactness. We provide a rigorous theoretical guarantee that maximizing the minimal sample margin over the entire dataset leads to maximizing the class margin regardless of feature dimension, class number, and class balancedness. It denotes that the sample margin also has the power of measuring inter-class separability.

According to the defined measures, we can obtain categorical discriminativeness of features by the loss function promoting the largest margins for both classes and samples, which also meets to tighten the margin-based generalization bound in (Kakade et al., 2008; Cao et al., 2019). The main contributions of this work are highlighted as follows:

- For a better understanding of margin-based losses, we provide a rigorous analysis about the necessity of normalization on prototypes and features. Moreover, we propose a generalized margin softmax loss (GM-Softmax), which can be derived to cover most of existing margin-based losses. We prove that, for the class-balance case, learning with the GM-Softmax loss leads to maximizing both class margin and sample margin under mild conditions.

- We show that learning with existing margin-based loss functions, such as SphereFace, NormFace, CosFace, AM-Softmax and ArcFace, would share the same optimal solution. In other words, all of them attempt to learn towards the largest margins, even though they are tailored to obtain different desired margins with explicit decision boundaries. However, these losses do not always maximize margins under different hyper-parameter settings. Instead, we propose an explicit *sample margin regularization* term and a novel *largest margin softmax loss* (LM-Softmax) derived from the minimal sample margin, which significantly improve the class margin and the sample margin.

- We consider the class-imbalanced case, in which the margins are severely affected. We provide a sufficient condition, which reveals that, if the centroid of prototypes is equal to zero, learning with GM-Softmax will provide the largest margins. Accordingly, we propose a simple but effective *zero-centroid regularization* term, which can be combined with commonly-used losses to mitigate class imbalance.

- Extensive experimental results are offered to demonstrate that the strategy of learning towards the largest margins significantly can improve the performance in accuracy and class/sample margins for various tasks, including visual classification, imbalanced classification, person re-identification, and face verification.

## 2 MEASURES OF INTRA-CLASS COMPACTNESS AND INTER-CLASS SEPARABILITY

With a labeled dataset $D = \{(\boldsymbol{x}_i, y_i)\}_{i=1}^{N}$ (where $\boldsymbol{x}_i$ denotes a training example with label $y_i$, and $y_i \in [1, k] = \{1, 2, ..., k\}$), the softmax loss for a $k$-classification problem is formulated as

$$L = \frac{1}{N} \sum_{i=1}^{N} - \log \frac{\exp(\boldsymbol{w}_{y_i}^{\mathrm{T}} \boldsymbol{z}_i)}{\sum_{j=1}^{k} \exp(\boldsymbol{w}_j^{\mathrm{T}} \boldsymbol{z}_i)} = \sum_{i=1}^{N} - \log \frac{\exp(\|\boldsymbol{w}_{y_i}\|_2 \|\boldsymbol{z}_i\|_2 \cos(\theta_{iy_i}))}{\sum_{j=1}^{k} \exp(\|\boldsymbol{w}_j\|_2 \|\boldsymbol{z}_i\|_2 \cos(\theta_{ij}))}, \quad (2.1)$$

where $z_i = \phi_\Theta(x_i) \in \mathbb{R}^d$ (usually $k \le d + 1$) is the learned feature representation vector; $\phi_\Theta$ denotes the feature extraction sub-network; $W = (w_1, ..., w_k) \in \mathbb{R}^{d \times k}$ denotes the linear classifier which is implemented with a linear layer at the end of the network (some works omit the bias and use an inner-product layer); $\theta_{ij}$ denotes the angle between $z_i$ and $w_j$; and $\|\cdot\|_2$ denotes the Euclidean norm, where $w_1, ..., w_k$ can be regarded as the class centers or prototypes (Mettes et al., 2019). For simplicity, we use prototypes to denote the weight vectors in the last inner-product layer.

The softmax loss intuitively encourages the learned feature representation $z_i$ to be similar to the corresponding prototype $w_{y_i}$, while pushing $z_i$ away from the other prototypes. Recently, some works (Liu et al., 2016; 2017; Deng et al., 2019) aim to achieve better performance by modifying the softmax loss with explicit decision boundaries to enforce extra intra-class compactness and inter-class separability. However, they do not provide the theoretical explanation and analysis about the newly designed losses. In this paper, we claim that a loss function to obtain better inter-class separability and intra-class compactness should learn towards the largest class and sample margin, and offer rigorously theoretical analysis as support. **All proofs can be found in the Appendix A.**

In the following, we define class margin and sample margin as the measures of inter-class separability and intra-class compactness, respectively, which serve as the base for our further derivation.

## 2.1 CLASS MARGIN

With prototypes $w_1, ..., w_k \in \mathbb{R}^d$, the class margin is defined as the minimal pairwise angle distance:

$$m_c(\{w_i\}_{i=1}^k) = \min_{i \ne j} \angle(w_i, w_j) = \arccos\left[\max_{i \ne j} \frac{w_i^T w_j}{\|w_i\|_2 \|w_j\|_2}\right], \quad (2.2)$$

where $\angle(w_i, w_j)$ denotes the angle between the vectors $w_i$ and $w_j$. Note that we omit the magnitudes of the prototypes in the definition, since the magnitudes tend to be very close according to the symmetry property. To verify this, we compute the ratio between the maximum and minimum magnitudes, which tends to be close to 1 on different datasets, as shown in Fig. 1.

To obtain better inter-class separability, we seek the largest class margin, which can be formulated as

$$\max_{\{w_i\}_{i=1}^k} m_c(\{w_i\}_{i=1}^k) = \max_{\{w_i\}_{i=1}^k} \min_{i \ne j} \angle(w_i, w_j).$$

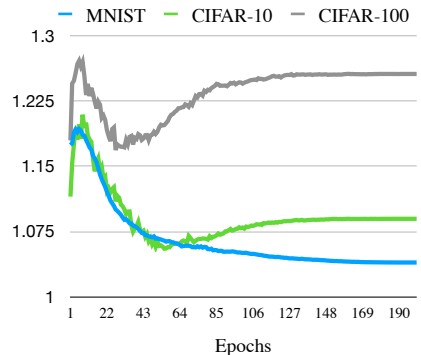

Figure 1: The curves of ratio between maximum and minimum magnitudes of prototypes on MNIST and CIFAR-10/-100 using the CE loss. The ratio is roughly close to 1 ($< 1.3$).

Since magnitudes do not affect the solution of the max-min problem, we perform $\ell_2$ normalization for each $w_i$ to effectively restrict the prototypes on the unit sphere $\mathbb{S}^{d-1}$ with center at the origin. Under this constraint, the maximization of the class margin is equivalent to the configuration of $k$ points on $\mathbb{S}^{d-1}$ to maximize their minimum pairwise distance:

$$\arg\max_{\{w_i\}_{i=1}^k \subset \mathbb{S}^{d-1}} \min_{i \ne j} \angle(w_i, w_j) = \arg\max_{\{w\}_{i=1}^k \subset \mathbb{S}^{d-1}} \min_{i \ne j} \|w_i - w_j\|_2, \quad (2.3)$$

The right-hand side is well known as the *k-points best-packing problem* on spheres (often called the *Tammes problem*), whose solution leads to the optimal separation of points (Borodachov et al., 2019). The best-packing problem turns out to be the limiting case of the minimal Riesz energy:

$$\arg\min_{\{w\}_{i=1}^k \subset \mathbb{S}^{d-1}} \lim_{t \to \infty} \sum_{i \ne j} \frac{1}{\|w_i - w_j\|_2^t} = \arg\max_{\{w\}_{i=1}^k \subset \mathbb{S}^{d-1}} \min_{i \ne j} \|w_i - w_j\|_2. \quad (2.4)$$

Interestingly, Liu et al. (2018) utilized the minimum hyperspherical energy as a generic regularization for neural networks to reduce undesired representation redundancy. When $w_1, ..., w_k \in \mathbb{S}^{d-1}$, $k \le d + 1$, and $t > 0$, the solution of the best-packing problem leads to the minimal Riesz $t$-energy:

**Lemma 2.1.** *For any $w_1, ..., w_k \in \mathbb{S}^{d-1}$, $d \ge 2$, and $2 \le k \le d + 1$, the solution of minimal Riesz $t$-energy and $k$-points best-packing configurations are uniquely given by the vertices of regular $(k-1)$-simplices inscribed in $\mathbb{S}^{d-1}$. Furthermore, $w_i^T w_j = \frac{-1}{k-1}, \forall i \ne j$.*

This lemma shows that the maximum of $m_c(\{\boldsymbol{w}_i\}_{i=1}^k)$ is $\arccos(\frac{-1}{k-1})$ when $k \leq d+1$, which is analytical and can be constructed artificially. However, when $k > d+1$, the optimal $k$-point configurations on the sphere $\mathbb{S}^{d-1}$ have no generic analytical solution, and are only known explicitly for a handful of cases, even for $d = 3$.

## 2.2 SAMPLE MARGIN

According to the definition in (Koltchinskii et al., 2002), for the network $\boldsymbol{f}(\boldsymbol{x}; \Theta, W) = W^{\mathrm{T}} \boldsymbol{\phi}_\Theta(\boldsymbol{x}) : \mathbb{R}^m \to \mathbb{R}^k$ that outputs $k$ logits, the sample margin for $(\boldsymbol{x}, y)$ is defined as

$$\gamma(\boldsymbol{x}, y) = \boldsymbol{f}(\boldsymbol{x})_y - \max_{j \neq y} \boldsymbol{f}(\boldsymbol{x})_j = \boldsymbol{w}_y^{\mathrm{T}} \boldsymbol{z} - \max_{j \neq y} \boldsymbol{w}_j^{\mathrm{T}} \boldsymbol{z}, \qquad (2.5)$$

where $\boldsymbol{z} = \boldsymbol{\phi}_\Theta(\boldsymbol{x})$ denotes the corresponding feature. Let $n_j$ be the number of samples in class $j$ and $S_j = \{i : y_i = j\}$ denote the sample indices corresponding to class $j$. We can further define the sample margin for samples in class $j$ as

$$\gamma_j = \min_{i \in S_j} \gamma(\boldsymbol{x}_i, y_i). \qquad (2.6)$$

Accordingly, the minimal sample margin over the entire dataset is $\gamma_{\min} = \min\{\gamma_1, ..., \gamma_k\}$. Intuitively, learning features and prototypes to maximize the minimum of all sample margins means making the feature embeddings close to their corresponding classes and far away from the others:

**Theorem 2.2.** *For $\boldsymbol{w}_1, ..., \boldsymbol{w}_k, \boldsymbol{z}_1, ..., \boldsymbol{z}_N \in \mathbb{S}^{d-1}$ (where $n_j > 0$ for each $j \in [1, k]$), the optimal solution $\{\boldsymbol{w}_i^*\}_{i=1}^k, \{\boldsymbol{z}_i^*\}_{i=1}^N = \arg\max_{\{\boldsymbol{w}_i\}_{i=1}^k, \{\boldsymbol{z}_i\}_{i=1}^N} \gamma_{\min}$ is obtained if and only if $\{\boldsymbol{w}_i^*\}_{i=1}^k$ maximizes the class margin $m_c(\{\boldsymbol{w}_i\}_{i=1}^k)$, and $\boldsymbol{z}_i^* = \frac{\boldsymbol{w}_{y_i}^* - \overline{\boldsymbol{w}}_{y_i}^*}{\|\boldsymbol{w}_{y_i}^* - \overline{\boldsymbol{w}}_{y_i}^*\|_2}$, where $\overline{\boldsymbol{w}}_{y_i}^*$ denotes the centroid of the vectors $\{\boldsymbol{w}_j : j \text{ maximizes } \boldsymbol{w}_{y_i}^{\mathrm{T}} \boldsymbol{w}_j, j \neq y_i\}$.*

As shown in the proof A, Theorem 2.2 guarantees that maximizing $\gamma_{\min}$ will provide the solution of the Tammes problem with respect to any feature dimension $d$, class number $k$, and both class-balanced and class-imbalance cases. When $2 \leq k \leq d+1$, we can derive the following proposition:

**Proposition 2.3.** *For any $\boldsymbol{w}_1, ..., \boldsymbol{w}_k, \boldsymbol{z}_1, ..., \boldsymbol{z}_N \in \mathbb{S}^{d-1}$, $d \geq 2$, and $2 \leq k \leq d+1$, the maximum of $\gamma_{\min}$ is $\frac{k}{k-1}$, which is obtained if and only if $\forall i \neq j$, $\boldsymbol{w}_i^{\mathrm{T}} \boldsymbol{w}_j = -\frac{1}{k-1}$, and $\boldsymbol{z}_i = \boldsymbol{w}_{y_i}$.*

Theorem 2.2 and Proposition 2.3 show that the best separation of prototypes is obtained when maximizing the minimal sample margin $\gamma_{\min}$.

On the other hand, let $L_{\gamma, j}[f] = \Pr_{\boldsymbol{x} \sim \mathcal{P}_j}[\max_{j' \neq j} f(\boldsymbol{x})_{j'} > f(\boldsymbol{x})_j - \gamma]$ denote the hard margin loss on samples from class $j$, and $\widehat{L}_{\gamma, j}$ denote its empirical variant. When the training dataset is separable (which indicates that there exists $f$ such that $\gamma_{\min} > 0$), Cao et al. (2019) provided a fine-grained generalization error bound under the setting with balanced test distribution by considering the margin of each class, *i.e.*, for $\gamma_j > 0$ and all $f \in \mathcal{F}$, with a high probability, we have

$$\Pr_{(\boldsymbol{x}, y)}[f(\boldsymbol{x})_y < \max_{l \neq y} f(\boldsymbol{x})_l] \leq \frac{1}{k} \sum_{j=1}^k \left( \widehat{L}_{\gamma_j, j}[f] + \frac{4}{\gamma_j} \widehat{\mathfrak{R}}_j(\mathcal{F}) + \varepsilon_j(\gamma_j) \right). \qquad (2.7)$$

In the right-hand side, the empirical Rademacher complexity $\frac{4}{\gamma_j} \widehat{\mathfrak{R}}_j(\mathcal{F})$ has a big impact. From the perspective of our work, a straightforward way to tighten the generalization bound is to enlarge the minimal sample margin $\gamma_{\min}$, which further leads to the larger margin $\gamma_j$ for each class $j$.

## 3 LEARNING TOWARDS THE LARGEST MARGINS

### 3.1 CLASS-BALANCED CASE

According to the above derivations, to encourage discriminative representation of features, the loss function should promote the largest possible margins for both classes and samples. In (Mettes et al., 2019), the pre-defined prototypes positioned through data-independent optimization are used to obtain a large class margin. As shown in Figure 2, although they keep the particularly large margin

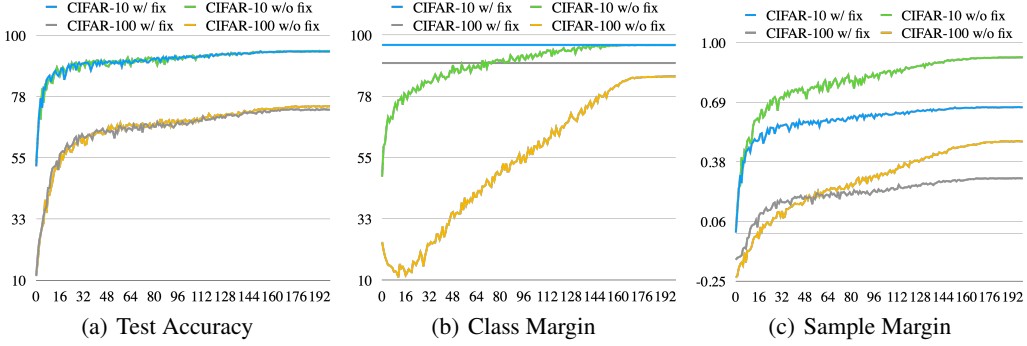

(a) Test Accuracy      (b) Class Margin      (c) Sample Margin

Figure 2: Test accuracies, class margins and sample margins on CIFAR-10 and CIFAR-100 with and without fixed prototypes, where fixed prototypes are pre-trained for very large class margins.

from the beginning, the sample margin is smaller than that optimized without fixed prototypes, leading to insignificant improvements in accuracy.

In recent years, in the design of variants of softmax loss, one popular approach (Bojanowski & Joulin, 2017; Wang et al., 2017; Mettes et al., 2019; Wang & Isola, 2020) is to perform normalization on prototypes or/and features, leading to superior performance than unnormalized counterparts (Parkhi et al., 2015; Schroff et al., 2015; Liu et al., 2017). However, there is no theoretical guarantee provided yet. In the following, we provide a rigorous analysis about the necessity of normalization. Firstly, we prove that minimizing the original softmax loss without normalization for both features and prototypes may result in a very small class margin:

**Theorem 3.1.** $\forall \varepsilon \in (0, \pi/2]$, *if the range of* $\boldsymbol{w}_1, ..., \boldsymbol{w}_k$ *or* $\boldsymbol{z}_1, ..., \boldsymbol{z}_N$ *is* $\mathbb{R}^d$ *($2 \leq k \leq d + 1$), then there exists prototypes that achieve the infimum of the softmax loss and have the class margin* $\varepsilon$.

This theorem reveals that, unless both features and prototypes are normalized, the original softmax loss may produce an arbitrary small class margin $\varepsilon$. As a corroboration of this conclusion, L-Softmax (Liu et al., 2016) and A-Softmax (Liu et al., 2017) that do not perform any normalization or only do on prototypes, cannot guarantee to maximize the class margin. To remedy this issue, some works (Wang et al., 2017; 2018a;b; Deng et al., 2019) proposed to normalize both features and prototypes.

A unified framework (Deng et al., 2019) that covers A-Softmax (Liu et al., 2017) with feature normalization, NormFace (Wang et al., 2017), CosFace/AM-Softmax (Wang et al., 2018b;a), ArcFace (Deng et al., 2019) as special cases can be formulated with hyper-parameters $m_1$, $m_2$ and $m_3$:

$$L'_i = -\log \frac{\exp(s(\cos(m_1\theta_{iy_i} + m_2) - m_3))}{\exp(s(\cos(m_1\theta_{iy_i} + m_2) - m_3)) + \sum_{j \neq y_i} \exp(s\cos\theta_{ij})}, \tag{3.1}$$

where $\theta_{ij} = \angle(\boldsymbol{w}_j, \boldsymbol{z}_i)$. The hyper-parameters setting usually guarantees that $\cos(m_1\theta_{iy_i} + m_2) - m_3 \leq \cos m_2 \cos \theta_{iy_i} - m_3$, and $m_2$ is usually set to satisfy $\cos m_2 \geq \frac{1}{2}$. Let $\alpha = \cos m_2$ and $\beta = -m_3 < 0$, then we have

$$L'_i \geq -\log \frac{\exp(s(\alpha\cos\theta_{iy_i} + \beta))}{\exp(s(\alpha\cos\theta_{iy_i} + \beta)) + \sum_{j \neq y_i} \exp(s\cos\theta_{ij})}, \tag{3.2}$$

which indicates that the existing well-designed normalized softmax loss functions are all considered as the upper bound of the right-hand side, and the equality holds if and only if $\theta_{iy_i} = 0$.

**Generalized Margin Softmax Loss.** Based on the right-hand side of (3.2), we can derive a more general formulation, called Generalized Margin Softmax (GM-Softmax) loss:

$$L_i = -\log \frac{\exp(s(\alpha_{i1}\cos\theta_{iy_i} + \beta_{i1}))}{\exp(s(\alpha_{i2}\cos\theta_{iy_i} + \beta_{i2})) + \sum_{j \neq y_i} \exp(s\cos\theta_{ij})}, \tag{3.3}$$

where $\alpha_{i1}, \alpha_{i2}, \beta_{i1}$ and $\beta_{i2}$ are hyper-parameters to handle the margins in training, which are set specifically for each sample instead of the same in (3.2). We also require that $\alpha_{i1} \geq \frac{1}{2}$, $\alpha_{i2} \leq \alpha_{i1}$, $s > 0$, $\beta_{i1}, \beta_{i2} \in \mathbb{R}$. For class-balanced case, each sample is treated equally, thus setting $\alpha_{i1} = \alpha_1$,

$\alpha_{i2} = \alpha_2$, $\beta_{i1} = \beta_1$ and $\beta_{i2} = \beta_2$, $\forall i$. For class-imbalanced case, the setting relies on the data distribution, *e.g.*, the LDAM loss (Cao et al., 2019) achieves the trade-off of margins with $\alpha_{i1} = \alpha_{i2} = 1$ and $\beta_{i1} = \beta_{i2} = -Cn_{y_i}^{-1/4}$. It is worth noting that we merely use the GM-Softmax loss as a theoretical formulation and will derive a more efficient form for the practical implementation.

Wang et al. (2017) provided a lower bound for normalized softmax loss, which relies on the assumption that all samples are well-separated, *i.e.*, each sample's feature is exactly the same as its corresponding prototype. However, this assumption could be invalid during training, *e.g.*, for binary classification, the best feature of the first class $z$ obtained by minimizing $-\log \frac{\exp(sw_1^T z)}{\exp(sw_1^T z) + \exp(sw_2^T z)}$ is $\frac{w_1 - w_2}{\|w_1 - w_2\|_2}$ rather than $w_1$. In the following, we provide a more general theorem, which does not rely on such a strong assumption. Moreover, we prove that the solutions $\{w_j^*\}_{j=1}^k$, $\{z_i^*\}_{i=1}^N$ minimizing the GM-Softmax loss will maximize both class margin and sample margin.

**Theorem 3.2.** *For class-balanced datasets, $w_1, ..., w_k, z_1, ..., z_N \in \mathbb{S}^{d-1}$, $d \geq 2$, and $2 \leq k \leq d + 1$, learning with GM-Softmax (where $\alpha_{i1} = \alpha_1$, $\alpha_{i2} = \alpha_2$, $\beta_{i1} = \beta_1$ and $\beta_{i2} = \beta_2$) leads to maximizing both class margin and sample margin.*

As can be seen, for any $\alpha_1 \geq \frac{1}{2}$, $\alpha_2 \leq \alpha_1$, $s > 0$, and $\beta_1, \beta \in \mathbb{R}$, minimizing the GM-Softmax loss produces the same optimal solution or leads to *neural collapse* (Papyan et al., 2020)), even though they are intuitively designed to obtain different decision boundaries. Moreover, we have

**Proposition 3.3.** *For class-balanced datasets, $w_1, ..., w_k, z_1, ..., z_N \in \mathbb{S}^{d-1}$, $d \geq 2$, and $2 \leq k \leq d + 1$, learning with the loss functions A-Softmax (Liu et al., 2017) with feature normalization, NormFace (Wang et al., 2017), CosFace (Wang et al., 2018b) or AM-Softmax (Wang et al., 2018a), and ArcFace (Deng et al., 2019) share the same optimal solution.*

Although these losses theoretically share the same optimal solution, in practice they usually meet sub-optimal solutions under different hyper-parameter settings when optimizing a neural network, which is demonstrated in Table 1. Moreover, these losses are complicated and possibly redundantly designed, leading to difficulties in practical implementation. Instead, we suggest a concise and easily implemented regularization term and a loss function in the following.

**Sample Margin Regularization.** In order to encourage learning towards the largest margins, we may explicitly leverage the sample margin (2.5) as the loss, which is defined as:

$$R_{sm}(x, y) = -(w_y^T z - \max_{j \neq y} w_j^T z). \qquad (3.4)$$

Noticeably, the empirical risk $\frac{1}{N}\sum_{i=1}^N R_{sm}(x_i, y_i)$ is a lower-bounded surrogate of $-\gamma_{min}$, *i.e.*, $-\gamma_{min} \geq \frac{1}{N}\sum_{i=1}^N R_{sm}(x_i, y_i)$, while directly minimizing $-\gamma_{min}$ is too difficult to optimize neural networks. When $k \leq d+1$, learning with $R_{sm}$ will promote the learning towards the largest margins:

**Theorem 3.4.** *For class-balanced datasets, $w_1, ..., w_k, z_1, ..., z_N \in \mathbb{S}^{d-1}$, $d \geq 2$, and $2 \leq k \leq d + 1$, learning with $R_{sm}$ leads to the maximization of both class margin and sample margin.*

Although learning with $R_{sm}$ theoretically achieves the largest margins, in practical implementation, the optimization by the gradient-based methods shows unstable and non-convergent results for large scale datasets. Alternatively, we turn to combine $R_{sm}$ as a regularization or complementary term with commonly-used losses, which is referred to as *sample margin regularization*. The empirical results demonstrate its superiority in learning towards the large margins, as depicted in Table 1.

**Largest Margin Softmax Loss (LM-Softmax).** Theorem 2.2 provides a theoretical guarantee that maximizing $\gamma_{min}$ will obtain the maximum of class margin regardless of feature dimension, class number, and class balancedness. It offers a straightforward approach to meet our purpose, *i.e.*, learning towards the largest margins. However, directly maximizing $\gamma_{min}$ is difficult to optimize a neural network with only one sample margin. As a consequence, we introduce a surrogate loss for balanced datasets, which is called the Largest Margin Softmax (LM-Softmax) loss:

$$L(x, y; s) = -\frac{1}{s}\log \frac{\exp(sw_y^T z)}{\sum_{j \neq y}\exp(sw_j^T z)} = \frac{1}{s}\log \sum_{j \neq y}\exp(s(w_j - w_y)^T z) \qquad (3.5)$$

which is derived by the limiting case of the logsumexp operator, *i.e.*. we have $-\gamma_{min} = \lim_{s \to \infty} \frac{1}{s}\log(\sum_{i=1}^N \sum_{j \neq y_i}\exp(s(w_j^T z_i - w_{y_i}^T z_i)))$. Moreover, since log is strictly concave, we

can derive the following inequality

$$\frac{1}{s}\log(\sum_{i=1}^{N}\sum_{j\neq y_i}\exp(s(\boldsymbol{w}_j^{\mathrm{T}}\boldsymbol{z}_i - \boldsymbol{w}_{y_i}^{\mathrm{T}}\boldsymbol{z}_i))) \geq \frac{1}{N}\sum_{i=1}^{N}L(\boldsymbol{x}_i, y_i; s) + \frac{1}{s}\log N. \tag{3.6}$$

Minimizing the right-hand side of (3.6) usually leads to that $\sum_{j\neq y_i}\exp(s(\boldsymbol{w}_j^{\mathrm{T}}\boldsymbol{z} - \boldsymbol{w}_{y_i}^{\mathrm{T}}\boldsymbol{z}))$ is a constant, while the equality of (3.6) holds if and only if $\sum_{j\neq y_i}\exp(s(\boldsymbol{w}_j^{\mathrm{T}}\boldsymbol{z} - \boldsymbol{w}_{y_i}^{\mathrm{T}}\boldsymbol{z}))$ is a constant. Thus, we can achieve the maximum of $\gamma_{\min}$ by minimizing $L(\boldsymbol{x}, y; s)$ defined in (3.5).

It can be found that, LM-Softmax can be regarded as a special case of the GM-Softmax loss when $\alpha_2$ or $\beta_2$ approaches $-\infty$, which can be more efficiently implemented than the GM-Softmax loss. With respect to the original softmax loss, LM-Softmax removes the term $\exp(s\boldsymbol{w}_y^{\mathrm{T}}\boldsymbol{z})$ in the denominator.

## 3.2    CLASS-IMBALANCED CASE

Class imbalance is ubiquitous and inherent in real-world classification problems (Buda et al., 2018; Liu et al., 2019). However, the performance of deep learning-based classification would drop significantly when the training dataset suffers from heavy class imbalance effect. According to (2.7), enlarging the sample margin can tighten the upper bound in case of class imbalance. To learn towards the largest margins on class-imbalanced datasets, we provide the following sufficient condition:

**Theorem 3.5.** *For class-balanced or -imbalanced datasets, $\boldsymbol{w}_1, ..., \boldsymbol{w}_k, \boldsymbol{z}_1, ..., \boldsymbol{z}_N \in \mathbb{S}^{d-1}$, $d \geq 2$, and $2 \leq k \leq d+1$, if $\sum_{i=1}^{K}\boldsymbol{w}_i = 0$, learning with GM-Softmax in (3.3) leads to maximizing both class margin and sample margin.*

This theorem reveals that, if the centroid of prototypes is equal to zero, learning with GM-Softmax will provide the largest margins.

**Zero-centroid Regularization.** As a consequence, we propose a straight regularization term as follows, which can be combined with commonly-used losses to remedy the class imbalance effect:

$$R_{\mathrm{w}}\{\boldsymbol{w}_j\}_{j=1}^{k} = \lambda\big\|\frac{1}{k}\sum_{j=1}^{k}\boldsymbol{w}_j\big\|_2^2. \tag{3.7}$$

The zero-centroid regularization is only applied to prototypes at the last inner-product layer.

## 4    EXPERIMENTS

In this section, we provide extensive experimental results to show superiority of our method on a variety of tasks, including visual classification, imbalanced classification, person ReID, and face verification. More experimental analysis and implementation details can be found in the appendix.

## 4.1    VISUAL CLASSIFICATION

To verify the effectiveness of the proposed sample margin regularization in improving inter-class separability and intra-class compactness, we conduct experiments of classification on balanced datasets MNIST (LeCun et al., 1998), CIFAR-10 and CIFAR-100 (Krizhevsky & Hinton, 2009). We evaluate performance with three metrics: 1) top-1 validation accuracy $acc$; 2) the class margin $m_{cls}$ defined in (2.2); 3) the average of sample margins $m_{samp}$. We use a 4-layer CNN, ResNet-18, and ResNet-34 on MNIST, CIFAR-10, and CIFAR-100, respectively. Moreover, some commonly-used neural units are considered, such as ReLU, BatchNorm, and cosine learning rate annealing. We use CE, CosFace, ArcFace, NormFace as the compared baseline methods. Note that CosFace, ArcFace, NormFace have one identical hyper-parameter $s$, which is used for comprehensive study.

**Results.** As shown in Table 1, all baseline losses fail in learning with large margins for all $s$, in which the class margin decreases as $s$ increases. There is no significant performance difference among them. In contrast, by coupling with the proposed sample margin regularization $R_{\mathrm{sm}}$, the losses turn to have larger margins. The results demonstrate that the proposed sample margin regularization is really beneficial to learn towards the possible largest margins. Moreover, the enlargement on class margin and sample margin means better inter-class separability and intra-class compactness, which further brings the improvement of classification accuracy in most cases.

Table 1: Test accuracies ($acc$), class margins ($m_{cls}$) and sample margins ($m_{samp}$) on MNIST, CIFAR-10 and CIFAR-100 using loss functions with/without $R_{sm}$ in (3.4). The results with positive gains are **highlighted**.

| Dataset | | MNIST | | | CIFAR-10 | | | CIFAR-100 | |
|---|---|---|---|---|---|---|---|---|---|
| Metric | $acc$ | $m_{cls}$ | $m_{samp}$ | $acc$ | $m_{cls}$ | $m_{samp}$ | $acc$ | $m_{cls}$ | $m_{samp}$ |
| CE | 99.11 | 87.39° | 0.5014 | 94.12 | 81.73° | 0.6203 | 74.56 | 65.38° | 0.1612 |
| CE + $0.5R_{sm}$ | **99.13** | **95.41°** | **1.026** | **94.45** | **96.31°** | **0.9744** | **74.96** | **90.00°** | **0.4955** |
| CosFace ($s = 10$) | 98.98 | 95.93° | 0.9839 | 94.39 | 96.00° | 0.9168 | 74.44 | 83.31° | 0.4578 |
| CosFace ($s = 20$) | 99.06 | 93.24° | 0.8376 | 94.13 | 91.22° | 0.7955 | 73.26 | 79.17° | 0.3078 |
| CosFace ($s = 64$) | 99.25 | 89.50° | 0.7581 | 93.53 | 64.14° | 0.6969 | 73.87 | 72.56° | 0.2233 |
| CosFace ($s = 10$) + $0.5R_{sm}$ | **99.16** | **95.56°** | **1.033** | **94.42** | **96.26°** | **0.9675** | 73.76 | **90.21°** | **0.5089** |
| CosFace ($s = 20$) + $0.5R_{sm}$ | **99.24** | **95.41°** | **1.030** | **94.27** | **96.18°** | **0.9490** | **74.41** | **89.02°** | **0.4780** |
| CosFace ($s = 64$) + $0.5R_{sm}$ | **99.27** | **95.35°** | **1.019** | **94.20** | **95.48°** | **0.9075** | **74.53** | **85.31°** | **0.3817** |
| ArcFace ($s = 10$) | 99.05 | 94.64° | 0.8225 | 94.50 | 91.23° | 0.8501 | 73.96 | 76.91° | 0.4313 |
| ArcFace ($s = 20$) | 99.11 | 90.84° | 0.6091 | 94.11 | 53.98° | 0.5707 | 74.74 | 60.91° | 0.3010 |
| ArcFace ($s = 64$) | 99.21 | 82.63° | 0.4038 | — | — | — | — | — | — |
| ArcFace ($s = 10$) + $0.5R_{sm}$ | **99.14** | **95.42°** | **1.034** | 94.21 | **96.27°** | **0.9651** | **74.47** | **90.13°** | **0.5143** |
| ArcFace ($s = 20$) + $0.5R_{sm}$ | **99.19** | **91.38°** | **1.030** | **94.32** | **96.15°** | **0.9571** | 74.64 | **88.73°** | **0.4804** |
| ArcFace ($s = 64$) + $0.5R_{sm}$ | 99.14 | **95.29°** | **1.019** | — | — | — | — | — | — |
| NormFace ($s = 10$) | 99.06 | 94.34° | 0.7750 | 94.16 | 94.40° | 0.8004 | 74.23 | 79.10° | 0.4250 |
| NormFace ($s = 20$) | 99.09 | 89.27° | 0.5263 | 94.09 | 74.32° | 0.6001 | 73.87 | 77.47° | 0.2498 |
| NormFace ($s = 64$) | 99.00 | 82.08° | 0.2621 | 94.01 | 36.50° | 0.2633 | 73.42 | 52.37° | 0.0993 |
| NormFace ($s = 10$) + $0.5R_{sm}$ | **99.16** | **95.38°** | **1.034** | **94.23** | **96.28°** | **0.9650** | **74.54** | **90.10°** | **0.5160** |
| NormFace ($s = 20$) + $0.5R_{sm}$ | **99.19** | **95.37°** | **1.031** | **94.38** | **96.17°** | **0.9519** | **74.75** | **88.86°** | **0.4773** |
| NormFace ($s = 64$) + $0.5R_{sm}$ | **99.34** | **95.29°** | **1.021** | **94.42** | **93.87°** | **0.9508** | 74.33 | **76.02°** | **0.3665** |

## 4.2 IMBALANCED CLASSIFICATION

To verify the effectiveness of the proposed zero-centroid regularization in handling class-imbalanced effect, we conduct experiments on imbalanced classification with two imbalance types: long-tailed imbalance (Cui et al., 2019) and step imbalance (Buda et al., 2018). The compared baseline losses include CE, Focal Loss, NormFace, CosFace, ArcFace, and the Label-Distribution-Aware Margin Loss (LDAM) with hyper-parameter $s = 5$. We follow the controllable data imbalance strategy in (Maas et al., 2011; Cao et al., 2019) to create the imbalanced CIFAR-10/-100 by reducing the number of training examples per class and keeping the validation set unchanged. The imbalance ratio $\rho = \max_i n_i / \min_i n_i$ is used to denote the ratio between sample sizes of the most frequent and least frequent classes. We add zero-centroid regularization to the margin-based baseline losses and the proposed LM-Softmax to verify its validity. We report the top-1 validation accuracy $acc$ and class margin $m_{cls}$ of compared methods.

Table 2: Test accuracies ($acc$) and class margins ($m_{cls}$) on imbalanced CIFAR-10. The results with positive gains are **highlighted** (where * denotes coupling with zero-centroid regularization term).

| Dataset | | | Imbalanced CIFAR-10 | | | | | | | Imbalanced CIFAR-100 | | | | | |
|---|---|---|---|---|---|---|---|---|---|---|---|---|---|---|---|
| Imbalance Type | | long-tailed | | | | step | | | | long-tailed | | | | step | |
| Imbalance Ratio | | 100 | | 10 | | 100 | | 10 | | 100 | | 10 | | 100 | | 10 |
| Metric | $acc$ | $m_{cls}$ | $acc$ | $m_{cls}$ | $acc$ | $m_{cls}$ | $acc$ | $m_{cls}$ | $acc$ | $m_{cls}$ | $acc$ | $m_{cls}$ | $acc$ | $m_{cls}$ | $acc$ | $m_{cls}$ |
| CE | 70.88 | 77.41° | 88.17 | 79.63° | 62.21 | 76.50° | 85.06 | 82.24° | 40.38 | 64.73° | 60.42 | 66.24° | 42.36 | 60.32° | 56.88 | 62.82° |
| Focal | 66.30 | 74.14° | 87.33 | 74.48° | 60.55 | 63.31° | 84.49 | 75.16° | 38.04 | 54.67° | 60.09 | 59.29° | 41.90 | 55.98° | 57.84 | 55.72° |
| CosFace | 69.28 | 58.77° | 87.02 | 81.61° | 53.64 | 19.78° | 84.86 | 75.96° | 34.91 | 4.731° | 60.60 | 70.81° | 40.36 | 0.764° | 47.56 | 8.559° |
| **CosFace*** | **69.52** | **91.90°** | **87.55** | **95.46°** | **62.49** | **95.86°** | **85.59** | **96.12°** | **40.98** | **80.93°** | **60.77** | **84.97°** | **41.17** | **41.59°** | **57.97** | **83.93°** |
| ArcFace | 72.20 | 65.86° | 89.00 | 85.23° | 62.48 | 54.29° | 86.32 | 80.51° | 42.77 | 13.22° | 63.21 | 67.73° | 41.47 | 0.497° | 58.89 | 0.369° |
| **ArcFace*** | **72.23** | **92.30°** | **89.22** | **96.23°** | **64.38** | **93.51°** | **86.65** | **96.23°** | **44.68** | **56.60°** | **63.80** | **73.45°** | **44.26** | **32.10°** | **60.79** | **79.85°** |
| NormFace | 72.37 | 62.72° | 89.19 | 82.60° | 63.69 | 51.00° | 86.37 | 77.82° | 43.71 | 16.11° | 63.50 | 71.26° | 41.93 | 1.363° | 59.85 | 21.32° |
| **NormFace*** | 72.07 | **94.95°** | **89.30** | **94.50°** | **64.07** | **93.06°** | **86.49** | **96.28°** | **44.25** | **64.85°** | **63.81** | **79.85°** | **44.51** | **36.30°** | **60.22** | **80.83°** |
| LDAM | 72.86 | 73.30° | 88.92 | 88.19° | 63.27 | 61.42° | 87.04 | 85.21° | 43.28 | 7.733° | 63.62 | 73.19° | 41.65 | 0.852° | 58.32 | 6.085° |
| **LDAM*** | 72.86 | **91.75°** | **89.51** | **96.26°** | **64.99** | **96.04°** | 86.74 | **96.26°** | **45.23** | **70.96°** | **64.18** | **85.03°** | **44.48** | **43.26°** | **60.83** | **75.22°** |
| LM-Softmax | 65.32 | 4.420° | 88.69 | 68.91° | 50.47 | 0.452° | 86.08 | 52.20° | 41.52 | 4.500° | 63.26 | 68.31° | 41.53 | 0.467° | 55.44 | 1.372° |
| **LM-Softmax*** | **73.21** | **92.57°** | **89.12** | **95.73°** | **65.91** | **93.84°** | **87.07** | **96.05°** | **45.28** | **69.53°** | **63.77** | **81.99°** | **46.23** | **43.15°** | **60.73** | **74.78°** |

**Results.** As can be seen from Table 2, the baseline margin-based losses have small class margins, although their classification performances are better than CE and Focal, which largely attribute to the normalization on feature and prototype. We can further improve their classification accuracy by

enlarging their class margins through the proposed zero-centroid regularization, as demonstrated by results in Table 2. Moreover, it can be found that the class margin of our LM-Softmax loss is fairly low in the severely imbalanced cases, since it is tailored for balanced case. We can also achieve significantly enlarged class margins and improved accuracy by the zero-centroid regularization.

## 4.3 PERSON RE-IDENTIFICATION

We conduct experiments on the task of person re-identification. Specifically, we use the off-the-shelf baseline (Luo et al., 2019) as the main code to verify the effectiveness of our proposed LM-Softmax. We follow the default parameter settings and training strategy, and train the ResNet50 with Triplet Loss Schroff et al. (2015) coupling with the compared losses, including the Softmax loss (CE), ArcFace, CosFace, NormFace, and our proposed LM-Softmax. Experiments are conducted on Market-1501 (Zheng et al., 2015) and DukeMTMC (Ristani et al., 2016). As shown in Table 3, our proposed LM-Softmax obtains obvious improvements in mean Average Precision (mAP), rank-1(Rank@1), and rank-5

Table 3: The results on Market-1501 and DukeMTMC for person re-identification task. The best three results are **highlighted**.

| Dataset | Market-1501 | | | DukeMTMC | | |
|---|---|---|---|---|---|---|
| Method | mAP | Rank1 | Rank@5 | mAP | Rank@1 | Rank@5 |
| CE | 82.8 | 92.7 | 97.5 | **73.0** | 83.5 | **93.0** |
| ArcFace ($s = 10$) | 67.5 | 84.1 | 92.1 | 37.7 | 58.7 | 72.7 |
| ArcFace ($s = 20$) | 79.1 | 90.8 | 96.5 | 61.4 | 78.3 | 88.6 |
| ArcFace ($s = 64$) | 80.4 | 92.6 | 97.4 | 67.6 | 83.4 | 91.4 |
| CosFace ($s = 10$) | 68.0 | 84.9 | 92.7 | 39.3 | 60.6 | 73.1 |
| CosFace ($s = 20$) | 80.5 | 92.0 | 97.1 | 64.2 | 81.3 | 89.7 |
| CosFace ($s = 64$) | 78.7 | 92.0 | 97.1 | 68.2 | 83.1 | 92.5 |
| NormFace ($s = 10$) | 81.2 | 91.6 | 96.3 | 63.7 | 79.3 | 88.5 |
| NormFace ($s = 20$) | 83.2 | **93.5** | **97.9** | 71.6 | 83.8 | 93.3 |
| NormFace ($s = 64$) | 77.5 | 90.0 | 96.9 | 60.1 | 75.2 | 88.1 |
| **LM-Softmax** ($s = 10$) | **83.3** | 92.8 | 97.1 | 72.2 | **85.8** | 92.4 |
| **LM-Softmax** ($s = 20$) | **84.7** | **93.8** | **97.6** | **74.1** | **86.4** | **93.5** |
| **LM-Softmax** ($s = 64$) | **84.6** | **93.9** | **98.1** | **74.2** | **86.6** | **93.5** |

(Rank@5) matching rate. Moreover, LM-Softmax exhibits significant robustness for different parameters, while ArcFace, CosFace, and NormFace show worse performance than ours and are more sensitive to parameter settings.

## 4.4 FACE VERIFICATION

We also verify our method on face verification that highly depends on the discriminability of feature embeddings. Following the settings in (An et al., 2020), we train the compared models on a large-scale dataset MS1MV3 (85K IDs/ 5.8M images) (Guo et al., 2016) and test on LFW (Huang et al., 2008), CFP-FP (Sengupta et al., 2016), AgeDB-30 (Moschoglou et al., 2017) and IJBC (Maze et al., 2018). We use ResNet34 as the backbone, and train it with batch size 512 for all compared methods. The comparison study includes CosFace, ArcFace, NormFace, and our LM-Softmax.

Table 4: Face verification results on IJBC-C, Age-DB30, CFP-FP and LFW. The results with positive gains are **highlighted**.

| Method | IJB-C | Age-DB30 | CFP-FP | LFW |
|---|---|---|---|---|
| ArcFace | 99.4919 | 98.067 | 97.371 | 99.800 |
| CosFace | 99.4942 | 98.033 | 97.300 | 99.800 |
| LM-Softmax | 99.4721 | 97.917 | 97.057 | 99.817 |
| ArcFace† | **99.5011** | **98.117** | **97.400** | **99.817** |
| ArcFace‡ | **99.5133** | **98.083** | **97.471** | **99.817** |
| CosFace† | **99.5112** | **98.150** | **97.371** | **99.817** |
| CosFace‡ | **99.5538** | 97.900 | **97.500** | 99.800 |
| LM-Softmax‡ | **99.5086** | **98.167** | **97.429** | **99.833** |

† and ‡ denotes training with $R_{sm}$ and $R_w$, respectively.

As shown in Table 4, $R_{sm}$ (sample margin regularization) and $R_w$ (zero-centroid regularization) can improve the performance of these baselines in most cases. Moreover, it is worth noting that the results of LM-Softmax are slightly worse than ArcFace and CosFace, which is due to that in these large-scale datasets there exists class imbalanced effect more or less. We can alleviate this issue by adding $R_w$, which can improve the performance further.

## 5 CONCLUSION

In this paper, we attempted to develop a principled mathematical framework for better understanding and design of margin-based loss functions, in contrast to the existing ones that are designed heuristically. Specifically, based on the class and sample margins, which are employed as measures of intra-class compactness and inter-class separability, we formulate the objective as learning towards the largest margins, and offer rigorously theoretical analysis as support. Following this principle, for class-balance case, we propose an explicit sample margin regularization term and a novel largest margin softmax loss; for class-imbalance case, we propose a simple but effective zero-centroid regularization term. Extensive experimental results demonstrate that the proposed strategy significantly improves the performance in accuracy and margins on various tasks.

**Acknowledgements.** This work was supported by National Key Research and Development Project under Grant 2019YFE0109600, National Natural Science Foundation of China under Grants 61922027, 6207115 and 61932022.

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

# Appendix for "Learning Towards the Largest Margin"

## A   PROOFS

**Lemma 2.1.** *For any $\boldsymbol{w}_1, ..., \boldsymbol{w}_k \in \mathbb{S}^{d-1}$, $d \geq 2$, and $2 \leq k \leq d+1$, the solution of minimal Riesz $t$-energy and $k$-points best-packing configurations are uniquely given by the vertices of regular $(k-1)$-simplices inscribed in $\mathbb{S}^{d-1}$. Furthermore, $\boldsymbol{w}_i^{\mathrm{T}}\boldsymbol{w}_j = \frac{-1}{k-1}, \forall i \neq j$.*

*Proof.* See in Borodachov et al. (2019, Theorem 3.3.1). □

**Theorem 2.2.** *For $\boldsymbol{w}_1, ..., \boldsymbol{w}_k, \boldsymbol{z}_1, ..., \boldsymbol{z}_N \in \mathbb{S}^{d-1}$ (where $n_j > 0$ for each $j \in [1, k]$), the optimal solution $\{\boldsymbol{w}_i^*\}_{i=1}^k, \{\boldsymbol{z}_i^*\}_{i=1}^N = \arg\max_{\{\boldsymbol{w}_i\}_{i=1}^k, \{\boldsymbol{z}_i\}_{i=1}^N} \gamma_{\min}$ is obtained if and only if $\{\boldsymbol{w}_i^*\}_{i=1}^k$ maximizes the class margin $m_c(\{\boldsymbol{w}_i\}_{i=1}^k)$, and $\boldsymbol{z}_i^* = \frac{\boldsymbol{w}_{y_i}^* - \overline{\boldsymbol{w}}_{y_i}^*}{\|\boldsymbol{w}_{y_i}^* - \overline{\boldsymbol{w}}_{y_i}^*\|_2}$, where $\overline{\boldsymbol{w}}_{y_i}^*$ denotes the centroid of the vectors $\{\boldsymbol{w}_j : j \text{ maximizes } \boldsymbol{w}_{y_i}^{\mathrm{T}}\boldsymbol{w}_j, j \neq y_i\}$.*

*Proof.* According to the definition of $\gamma_{\min}$, we have

$$
\begin{aligned}
\arg\max_{\boldsymbol{w}} \max_{\boldsymbol{z}} \gamma_{\min} &= \arg\max_{\boldsymbol{w}} \max_{\boldsymbol{z}} \min_i \boldsymbol{w}_{y_i}^{\mathrm{T}}\boldsymbol{z}_i - \max_{j \neq y_i} \boldsymbol{w}_j^{\mathrm{T}}\boldsymbol{z}_i \\
&= \arg\max_{\boldsymbol{w}} \min_i \max_{\boldsymbol{z}_i} \boldsymbol{w}_{y_i}^{\mathrm{T}}\boldsymbol{z}_i - \max_{j \neq y_i} \boldsymbol{w}_j^{\mathrm{T}}\boldsymbol{z}_i \\
&= \arg\max_{\boldsymbol{w}} \min_i \max_{\boldsymbol{z}_i} \boldsymbol{w}_{y_i}^{\mathrm{T}}\boldsymbol{z}_i - \boldsymbol{w}_k^{\mathrm{T}}\boldsymbol{z}_i \\
&= \arg\max_{\boldsymbol{w}} \min_i \|\boldsymbol{w}_{y_i} - \boldsymbol{w}_k\|_2
\end{aligned}
,
$$

where $k = \arg\max_{j \neq y_i} \boldsymbol{w}_j^{\mathrm{T}}\boldsymbol{z}_i$, and $\boldsymbol{z}_i = \frac{\boldsymbol{w}_{y_i} - \boldsymbol{w}_k}{\|\boldsymbol{w}_{y_i} - \boldsymbol{w}_k\|_2}$. Notice that $\boldsymbol{w}_k^{\mathrm{T}}\boldsymbol{z}_i = -\frac{1}{2}\|\boldsymbol{w}_{y_i} - \boldsymbol{w}_k\|_2$, then $k = \arg\min_{j \neq y_i} \|\boldsymbol{w}_{y_i}^{\mathrm{T}} - \boldsymbol{w}_j\|_2$. Therefore, we have

$$
\arg\max_{\boldsymbol{w}} \max_{\boldsymbol{z}} \gamma_{\min} = \max_{\boldsymbol{w}} \min_i \min_{k \neq y_i} \|\boldsymbol{w}_{y_i} - \boldsymbol{w}_k\|_2 = \arg\max_{\boldsymbol{w}} \min_{i \neq j} \|\boldsymbol{w}_i - \boldsymbol{w}_j\|_2,
$$

i.e., maximizing $\gamma_{\min}$ will provide the solution of the Tammes Problem, which also maximizes the class margin.

On the other hand, $\boldsymbol{z}_i^*$ maximizes $\boldsymbol{w}_{y_i}^{*\mathrm{T}}\boldsymbol{z}_i - \max_{j \neq y_i} \boldsymbol{w}_j^{*\mathrm{T}}\boldsymbol{z}_i$, i.e.,

$$
\begin{aligned}
\boldsymbol{z}_i^* &= \arg\max_{\boldsymbol{z}_i \in \mathbb{S}^{d-1}} \boldsymbol{w}_{y_i}^{*\mathrm{T}}\boldsymbol{z}_i - \max_{j \neq y_i} \boldsymbol{w}_j^{*\mathrm{T}}\boldsymbol{z}_i \\
&= \arg\max_{\boldsymbol{z}_i \in \mathbb{S}^{d-1}} \boldsymbol{w}_{y_i}^{*\mathrm{T}}\boldsymbol{z}_i - \overline{\boldsymbol{w}}_{y_i}^{*\mathrm{T}}\boldsymbol{z}_i \\
&= \frac{\boldsymbol{w}_{y_i}^* - \overline{\boldsymbol{w}}_{y_i}^*}{\|\boldsymbol{w}_{y_i}^* - \overline{\boldsymbol{w}}_{y_i}^*\|_2}
\end{aligned}
,
$$

where $\overline{\boldsymbol{w}}_{y_i}^*$ denotes the centroid of the vectors $\{\boldsymbol{w}_j : j \text{ maximizes } \boldsymbol{w}_{y_i}^{\mathrm{T}}\boldsymbol{w}_j, j \neq y_i\}$. □

**Proposition 2.3.** *For any $\boldsymbol{w}_1, ..., \boldsymbol{w}_k, \boldsymbol{z}_1, ..., \boldsymbol{z}_N \in \mathbb{S}^{d-1}$, $d \geq 2$, and $2 \leq k \leq d+1$, the maximum of $\gamma_{\min}$ is $\frac{k}{k-1}$, which is obtained if and only if $\forall i \neq j$, $\boldsymbol{w}_i^{\mathrm{T}}\boldsymbol{w}_j = -\frac{1}{k-1}$, and $\boldsymbol{z}_i = \boldsymbol{w}_{y_i}$.*

*Proof.* Based on Theorem 2.2, the maximum of $\gamma_{\min}$ is obtained if and only if $\{\boldsymbol{w}_i\}_{i=1}^k$ maximizes the class margin and $\boldsymbol{z}_i = \frac{\boldsymbol{w}_{y_i} - \overline{\boldsymbol{w}}_{y_i}}{\|\boldsymbol{w}_{y_i} - \overline{\boldsymbol{w}}_{y_i}\|_2}$, i.e., $\boldsymbol{w}_i^{\mathrm{T}}\boldsymbol{w}_j = -\frac{1}{k-1}$ according to Lemma 2.3. At this time, we have $\boldsymbol{z}_i = \frac{\boldsymbol{w}_{y_i} - \overline{\boldsymbol{w}}_{y_i}}{\|\boldsymbol{w}_{y_i} - \overline{\boldsymbol{w}}_{y_i}\|_2} = \frac{\boldsymbol{w}_{y_i} - (-\boldsymbol{w}_{y_i})}{\|\boldsymbol{w}_{y_i} - (-\boldsymbol{w}_{y_i})\|_2} = \boldsymbol{w}_{y_i}$. □

**Theorem 3.1.** *$\forall \varepsilon \in (0, \pi/2]$, if the range of $\boldsymbol{w}_1, ..., \boldsymbol{w}_k$ or $\boldsymbol{z}_1, ..., \boldsymbol{z}_N$ is $\mathbb{R}^d$ ($2 \leq k \leq d+1$), then there exists prototypes that achieve the infimum of the softmax loss and have the class margin $\varepsilon$.*

*Proof.* With the softmax loss, the goal is to optimize the following problem

$$\min_{\{\boldsymbol{w}_j\}_{j=1}^k, \{\boldsymbol{z}_i\}_{i=1}^N} L = -\frac{1}{N} \sum_{i=1}^N \log \frac{\exp(\boldsymbol{w}_{y_i}^{\mathrm{T}} \boldsymbol{z}_i)}{\sum_{j=1}^K \exp(\boldsymbol{w}_j^{\mathrm{T}} \boldsymbol{z}_i)}.$$

$\forall \varepsilon \in (0, \frac{\pi}{2}]$, we can easily obtain $k \leq d+1$ vectors $\boldsymbol{w}_1', ..., \boldsymbol{w}_k'$ on the unit sphere $\mathbb{S}^{d-1}$, such that the angle between any two of them is $\varepsilon \in (0, \frac{\pi}{2})$.

(1) If the domain of $\boldsymbol{w}_1, ..., \boldsymbol{w}_k$ is $\mathbb{R}^d$, then let $\boldsymbol{w}_j = s\boldsymbol{w}_j'$, and $\boldsymbol{z}_i = \boldsymbol{w}_{y_i}$. In this way, we have $\boldsymbol{w}_{y_i}^{\mathrm{T}} \boldsymbol{z}_i > \boldsymbol{w}_j^{\mathrm{T}} \boldsymbol{z}_i, \forall j \neq y_i$. The infimum of softmax loss can be obtained by directly increasing $s$.

(2) If the domain of $\boldsymbol{z}_1, ..., \boldsymbol{z}_k$ is $\mathbb{R}^d$, then let $\boldsymbol{w}_j = \boldsymbol{w}_j'$, and $\boldsymbol{z}_i = s\boldsymbol{w}_{y_i}$. In this way, we have $\boldsymbol{w}_{y_i}^{\mathrm{T}} \boldsymbol{z}_i > \boldsymbol{w}_j^{\mathrm{T}} \boldsymbol{z}_i, \forall j \neq y_i$. The infimum of softmax loss can be obtained by directly increasing $s$.

In conclusion, without both normalization for both features and prototypes, the original softmax loss may produce an arbitrary small class margin $\varepsilon$. $\square$

**Theorem 3.2.** *For class-balanced datasets (i.e., each class has the same number of samples), $\boldsymbol{w}_1, ..., \boldsymbol{w}_k, \boldsymbol{z}_1, ..., \boldsymbol{z}_N \in \mathbb{S}^{d-1}$, $d \geq 2$, and $2 \leq k \leq d+1$, learning with GM-Softmax (where $\alpha_{i1} = \alpha_1, \alpha_{i2} = \alpha_2, \beta_{i1} = \beta_1$ and $\beta_{i2} = \beta_2$) leads to maximizing both the class margin and the sample margin. More specifically, the optimal solution*

$$\{\boldsymbol{w}_j^*\}_{j=1}^k, \{\boldsymbol{z}_i^*\}_{i=1}^N = \arg\min_{\boldsymbol{w}_j, \boldsymbol{z}_i \in \mathbb{S}^{d-1}} \frac{1}{N} \sum_{i=1}^N -\log \frac{\exp(s(\alpha_1 \boldsymbol{w}_{y_i}^{\mathrm{T}} \boldsymbol{z}_i + \beta_1))}{\exp(s(\alpha_2 \boldsymbol{w}_{y_i}^{\mathrm{T}} \boldsymbol{z}_i + \beta_2)) + \sum_{j \neq y_i} \exp(s\boldsymbol{w}_j^{\mathrm{T}} \boldsymbol{z}_i)}$$

*have the largest class margin $m_c^* = \arccos \frac{-1}{k-1}$ and the largest sample margin $\gamma_{\min}^* = \frac{k}{k-1}$. The lower bound of the risk is $\log[\exp(s(\alpha_1 + \beta_1 - \alpha_2 - \beta_2)) + (k-1)\exp(-s(\frac{1}{k-1} + \alpha_1 + \beta_1))]$, which is obtained if and only if $\forall i \neq j$, $\boldsymbol{w}_i^{\mathrm{T}} \boldsymbol{w}_j = \frac{-1}{k-1}$, and $\boldsymbol{z}_i = \boldsymbol{w}_{y_i}$.*

*Proof.* Since the function exp is strictly convex, using the Jensen's inequality, we have

$$L = \frac{1}{N} \sum_{i=1}^N -\log \frac{\exp(s(\alpha_1 \boldsymbol{w}_{y_i}^{\mathrm{T}} \boldsymbol{z}_i + \beta_1))}{\exp(s(\alpha_2 \boldsymbol{w}_{y_i}^{\mathrm{T}} \boldsymbol{z}_i + \beta_2)) + \sum_{j \neq i} \exp(s\boldsymbol{w}_j^{\mathrm{T}} \boldsymbol{z}_i)}$$

$$\geq \frac{1}{N} \sum_{i=1}^N -\log \frac{\exp(s(\alpha_1 \boldsymbol{w}_{y_i}^{\mathrm{T}} \boldsymbol{z}_i + \beta_1))}{\exp(s(\alpha_2 \boldsymbol{w}_{y_i}^{\mathrm{T}} \boldsymbol{z}_i + \beta_2)) + (k-1)\exp(\frac{s}{k-1} \sum_{j \neq i} \boldsymbol{w}_j^{\mathrm{T}} \boldsymbol{z}_i)}$$

Let $\overline{\boldsymbol{w}} = \frac{1}{k} \sum_{i=1}^k \boldsymbol{w}_i$, $\alpha = \alpha_2 - \alpha_1$, $\beta = \beta_2 - \beta_1$, $\sigma = \frac{k}{k-1}$, and $\delta = \frac{1}{k-1} + \alpha_1$, then we have

$$L \geq \frac{1}{N} \sum_{i=1}^N \log\left[\exp(s(\alpha \boldsymbol{w}_{y_i}^{\mathrm{T}} \boldsymbol{z}_i + \beta) + (k-1)\exp(s(\sigma\overline{\boldsymbol{w}} - \delta\boldsymbol{w}_{y_i})^{\mathrm{T}} \boldsymbol{z}_i - s\beta_1)\right]$$

$$\geq \frac{1}{N} \sum_{i=1}^N \log[\exp(s\alpha + s\beta) + (k-1)\exp(-s\|\sigma\overline{\boldsymbol{w}} - \delta\boldsymbol{w}_{y_i}\|_2 - s\beta_1)] \qquad ,$$

$$= \frac{1}{k} \sum_{i=1}^k \log[\exp(s\alpha + s\beta) + (k-1)\exp(-s\|\sigma\overline{\boldsymbol{w}} - \delta\boldsymbol{w}_i\|_2 - s\beta_1)]$$

where we use the facts that $\alpha \boldsymbol{w}_{y_i}^{\mathrm{T}} \boldsymbol{z}_i \geq \alpha$ when $\alpha \leq 0$, $(\sigma\overline{\boldsymbol{w}} - \delta\boldsymbol{w}_{y_i})^{\mathrm{T}} \boldsymbol{z}_i \geq -\|\sigma\overline{\boldsymbol{w}} - \delta\boldsymbol{w}_i\|_2$ when $\boldsymbol{z}_i \in \mathbb{S}^{d-1}$. Due the convexity of the function $\log[1 + \exp(ax + b)]$ $(a > 0)$, we use the Jensen's

inequality and obtain that

$$
\begin{aligned}
L &\geq \log\left[\exp(s(\alpha+\beta)) + (k-1)\exp(-\frac{s}{k}\sum_{i=1}^{k}\|\sigma\overline{\boldsymbol{w}} - \delta\boldsymbol{w}_i\|_2 - s\beta_1)\right] \\
&\geq \log\left[\exp(s(\alpha+\beta)) + (k-1)\exp(-\frac{s}{k}\sqrt{k\sum_{i=1}^{k}\|\sigma\overline{\boldsymbol{w}} - \delta\boldsymbol{w}_i\|_2^2} - s\beta_1)\right] \\
&= \log\left[\exp(s(\alpha+\beta)) + (k-1)\exp(-\frac{s}{k}\sqrt{k(k\delta^2 - 2k\sigma\delta\|\overline{\boldsymbol{w}}\|_2^2 + k\sigma^2\|\overline{\boldsymbol{w}}\|_2^2)} - s\beta_1)\right], \\
&\geq \log[\exp(s(\alpha+\beta)) + (k-1)\exp(-s(\delta+\beta_1))] \\
&= \log\left[\exp(s(\alpha_2 - \alpha_1 + \beta_2 - \beta_1)) + (k-1)\exp(-s(\frac{1}{k-1}+\alpha_1+\beta_1))\right]
\end{aligned}
$$

where in the second inequality we used the Cauchy–Schwarz inequality, and the third inequality is based on that $\sigma \leq 2\delta \Leftrightarrow \alpha_1 \geq \frac{k-2}{2k-2}$, which holds since $\alpha_1 \geq \frac{1}{2}$.

According to the above derivation, the equality holds if and only if $\forall i$, $\boldsymbol{w}_1^{\mathrm{T}}\boldsymbol{z}_i = ... = \boldsymbol{w}_{y_i-1}^{\mathrm{T}}\boldsymbol{z}_i = \boldsymbol{w}_{y_i+1}^{\mathrm{T}}\boldsymbol{z}_i = ... = \boldsymbol{w}_k^{\mathrm{T}}\boldsymbol{z}_i$, $\boldsymbol{w}_{y_i}^{\mathrm{T}}\boldsymbol{z}_i = 1$, $\boldsymbol{z}_i = -\frac{\sigma\overline{\boldsymbol{w}} - \delta\boldsymbol{w}_{y_i}}{\|\sigma\overline{\boldsymbol{w}} - \delta\boldsymbol{w}_{y_i}\|_2}$, $\|\sigma\overline{\boldsymbol{w}} - \delta\boldsymbol{w}_1\|_2 = ... = \|\sigma\overline{\boldsymbol{w}} - \delta\boldsymbol{w}_k\|_2$, and $\overline{\boldsymbol{w}} = 0$. The condition can be simplified as $\forall i \neq j$, $\boldsymbol{w}_i^{\mathrm{T}}\boldsymbol{w}_j = \frac{-1}{k-1}$, and $\boldsymbol{z}_i = \boldsymbol{w}_{y_i}$ when $2 \leq d$ and $2 \leq k \leq d+1$. $\qquad\square$

**Proposition 3.3.** *For class-balanced datasets, $\boldsymbol{w}_1, ..., \boldsymbol{w}_k, \boldsymbol{z}_1, ..., \boldsymbol{z}_N \in \mathbb{S}^{d-1}$, $d \geq 2$, and $2 \leq k \leq d+1$, learning with the loss functions A-Softmax (Liu et al., 2017) with feature normalization, NormFace (Wang et al., 2017), CosFace (Wang et al., 2018b) or AM-Softmax (Wang et al., 2018a), and ArcFace (Deng et al., 2019) share the same optimal solution.*

*Proof.* A unified framework for A-Softmax with feature normalization, NormFace, LMLC/AM-Softmax and ArcFace can be implemented with hyper-parameters $m_1$, $m_2$ and $m_3$, *i.e.*,

$$
L_i' = -\log\frac{\exp(s(\cos(m_1\theta_{iy_i} + m_2) - m_3))}{\exp(s(\cos(m_1\theta_{iy_i} + m_2) - m_3)) + \sum_{j \neq y_i}\exp(s\cos\theta_{ij})},
$$

where $\theta_{ij} = \angle(\boldsymbol{w}_j, \boldsymbol{z}_i)$. The setting of these hyper-parameters always guarantees that $\cos(m_1\theta_{iy_i} + m_2) - m_3 \leq \cos m_2 \cos\theta_{iy_i} - m_3$, and $m_2$ is usually set to satisfy $\cos m_2 \geq \frac{1}{2}$. Let $\alpha = \cos m_2$ and $\beta = -m_3 < 0$, then we have

$$
L_i' \geq -\log\frac{\exp(s(\alpha\cos\theta_{iy_i} + \beta))}{\exp(s(\alpha\cos\theta_{iy_i} + \beta)) + \sum_{j \neq y_i}\exp(s\cos\theta_{ij})}, \tag{A.1}
$$

where the equality holds if and only if $\theta_{iy_i} = 0$.

According to Theorem 3.2, we know that the empirical risk of the loss function in the right-hand side of (3.2) has a lower bound, then we obtain

$$
\frac{1}{N}\sum_{i=1}^{N}L_i' \geq -\log\frac{\exp(s(\alpha+\beta))}{\exp(s(\alpha+\beta)) + \sum_{j \neq y_i}\exp(-\frac{s}{k-1})} \tag{A.2}
$$

The equality holds if and only if $\forall i \neq j$, $\boldsymbol{w}_i^{\mathrm{T}}\boldsymbol{w}_j = \frac{-1}{k-1}$, and $\boldsymbol{z}_i = \boldsymbol{w}_{y_i}$. Since $\boldsymbol{z}_i = \boldsymbol{w}_{y_i}$ means $\theta_{iy_i} = 0$, indicating that the equality in (3.2) holds. Then the optimal solution is the same for A-Softmax with feature normalization, NormFace, CosFace, and ArcFace. $\qquad\square$

**Theorem 3.4.** *For class-balanced datasets, $\boldsymbol{w}_1, ..., \boldsymbol{w}_k, \boldsymbol{z}_1, ..., \boldsymbol{z}_N \in \mathbb{S}^{d-1}$, $d \geq 2$, and $2 \leq k \leq d+1$, learning with $R_{sm} = -\boldsymbol{w}_y^{\mathrm{T}}\boldsymbol{z} + \max_{j \neq y}\boldsymbol{w}_j^{\mathrm{T}}\boldsymbol{z}$ leads to the maximization of the class margin and the sample margin.*

*Proof.* Let $L(\boldsymbol{z}, y) = -\boldsymbol{w}_y^{\mathrm{T}}\boldsymbol{z} + \max_{j \neq y}\boldsymbol{w}_j^{\mathrm{T}}\boldsymbol{z}$, $\overline{\boldsymbol{w}} = \frac{1}{k}\sum_{i=1}^{k}\boldsymbol{w}_i$, then we have

$$
\begin{aligned}
\frac{1}{N}\sum_{i=1}^{N} L(z_i, y_i) &= \frac{1}{N}\sum_{i=1}^{N}(-\boldsymbol{w}_{y_i}^{\mathrm{T}}\boldsymbol{z}_i + \max_{j \neq y_i}\boldsymbol{w}_j^{\mathrm{T}}\boldsymbol{z}_i) \\
&\geq \frac{1}{N}\sum_{i=1}^{N}(-\boldsymbol{w}_{y_i}^{\mathrm{T}}\boldsymbol{z}_i + \frac{1}{k-1}\sum_{j \neq y_i}\boldsymbol{w}_j^{\mathrm{T}}\boldsymbol{z}_i) \\
&= \frac{k}{N(k-1)}\sum_{i=1}^{N} -(\boldsymbol{w}_{y_i} - \overline{\boldsymbol{w}})^{\mathrm{T}}\boldsymbol{z}_i \\
&\geq \frac{k}{N(k-1)}\sum_{i=1}^{N} -\|\boldsymbol{w}_{y_i} - \overline{\boldsymbol{w}}\|_2 \\
&= \frac{1}{k-1}\sum_{i=1}^{k} -\|\boldsymbol{w}_i - \overline{\boldsymbol{w}}\|_2 \\
&\geq \frac{1}{k-1} - \sqrt{k(\sum_{i=1}^{k}\|\boldsymbol{w}_i - \overline{\boldsymbol{w}}\|_2^2)} \\
&= \frac{1}{k-1} - \sqrt{k(k - k\|\overline{\boldsymbol{w}}\|_2^2)} \\
&\geq -\frac{k}{k-1}
\end{aligned}
\tag{A.3}
$$

where the equality holds if and only if $\forall i \neq j$, $\boldsymbol{w}_i^{\mathrm{T}}\boldsymbol{w}_j = \frac{-1}{k-1}$, and $\boldsymbol{z}_i = \boldsymbol{w}_{y_i}$. $\qquad\square$

**Theorem 3.5.** *For class-balanced or -imbalanced cases, $\boldsymbol{w}_1, ..., \boldsymbol{w}_k, \boldsymbol{z}_1, ..., \boldsymbol{z}_N \in \mathbb{S}^{d-1}$, $d \geq 2$, and $2 \leq k \leq d + 1$, if $\sum_{i=1}^{K}\boldsymbol{w}_i = 0$, then learning with the GM-Softmax loss in (3.3) leads to maximizing both the class margin and the sample margin. More specifically, the optimal solution $\{\boldsymbol{w}_j^*\}_{j=1}^{K}, \{\boldsymbol{z}_i^*\}_{i=1}^{N}$ has the largest class margin $m(\boldsymbol{W}^*) = \arccos\frac{-1}{K-1}$ and the largest sample margin $\gamma_{\min}^* = \frac{k}{k-1}$. The lower bound of the risk is $\frac{1}{N}\sum_{i=1}^{N}\log[\exp(s(\alpha_{i1} + \beta_{i1} - \alpha_{i2} - \beta_{i2})) + (k-1)\exp(-s(\frac{1}{k-1} + \alpha_{i1} + \beta_{i1}))]$, which is obtained if and only if $\forall i \neq j$, $\boldsymbol{w}_i^{\mathrm{T}}\boldsymbol{w}_j = \frac{-1}{K-1}$, and $\boldsymbol{z}_i = \boldsymbol{w}_{y_i}$, i.e., the optimal solution maximizes that class margin and sample margin.*

*Proof.* For the GM-Softmax loss $L_i = -\log\frac{\exp(s(\alpha_{i1}\boldsymbol{w}_{y_i}^{\mathrm{T}}\boldsymbol{z}_i + \beta_{i1}))}{\exp(s(\alpha_{i2}\boldsymbol{w}_{y_i}^{\mathrm{T}}\boldsymbol{z}_i + \beta_{i2})) + \sum_{j \neq y_i}\exp(s\boldsymbol{w}_j^{\mathrm{T}}\boldsymbol{z}_i)}$, let $\alpha_i = \alpha_{i2} - \alpha_{i1} \leq 0$, $\beta_i = \beta_{i2} - \beta_{i1}$. If $\sum_{i=1}^{K}\boldsymbol{w}_i = 0$, then we have

$$
\begin{aligned}
L_i &= -\log\frac{\exp(s(\alpha_{i1}\boldsymbol{w}_{y_i}^{\mathrm{T}}\boldsymbol{z}_i + \beta_{i1}))}{\exp(s(\alpha_{i2}\boldsymbol{w}_{y_i}^{\mathrm{T}}\boldsymbol{z}_i + \beta_{i2})) + \sum_{j \neq y_i}\exp(s\boldsymbol{w}_j^{\mathrm{T}}\boldsymbol{z}_i)} \\
&\geq -\log\frac{\exp(s(\alpha_{i1}\boldsymbol{w}_{y_i}^{\mathrm{T}}\boldsymbol{z}_i + \beta_{i1}))}{\exp(s(\alpha_{i2}\boldsymbol{w}_{y_i}^{\mathrm{T}}\boldsymbol{z}_i + \beta_{i2})) + (k-1)\exp(\frac{1}{k-1}\sum_{j \neq y_i}s\boldsymbol{w}_j^{\mathrm{T}}\boldsymbol{z}_i)} \\
&= -\log\frac{\exp(s(\alpha_{i1}\boldsymbol{w}_{y_i}^{\mathrm{T}}\boldsymbol{z}_i + \beta_{i1}))}{\exp(s(\alpha_{i2}\boldsymbol{w}_{y_i}^{\mathrm{T}}\boldsymbol{z}_i + \beta_{i2})) + (k-1)\exp(-\frac{s}{k-1}\boldsymbol{w}_{y_i}^{\mathrm{T}}\boldsymbol{z}_i)} \\
&= \log\left[\exp(s\alpha_i\boldsymbol{w}_{y_i}^{\mathrm{T}}\boldsymbol{z}_i + s\beta_i) + (k-1)\exp(-\frac{s}{k-1}\boldsymbol{w}_{y_i}^{\mathrm{T}}\boldsymbol{z}_i - s(\alpha_{i1}\boldsymbol{w}_{y_i}^{\mathrm{T}}\boldsymbol{z}_i + \beta_{i1}))\right] \\
&\geq \log\left[\exp(s(\alpha_{i1} + \beta_{i1} - \alpha_{i2} - \beta_{i2})) + (k-1)\exp(-s(1/(k-1) + \alpha_{i1} + \beta_{i1}))\right]
\end{aligned}
\tag{A.4}
$$

where in the first inequality we used the Jensen's inequality, and the last inequality comes from the facts that $\alpha_i\boldsymbol{w}_{y_i}^{\mathrm{T}}\boldsymbol{z}_i \geq \alpha_i$ and $-\frac{1}{k-1}\boldsymbol{w}_{y_i}^{\mathrm{T}}\boldsymbol{z}_i - \alpha_{i1}\boldsymbol{w}_{y_i}^{\mathrm{T}}\boldsymbol{z}_i \geq -\frac{1}{k-1} - \alpha_{i1}$.

Therefore, we have the lower bound of the risk $\frac{1}{N}\sum_{i=1}^{N} L_i \geq \frac{1}{N}\sum_{i=1}^{N}\log[\exp(s(\alpha_{i1} + \beta_{i1} - \alpha_{i2} - \beta_{i2})) + (k-1)\exp(-s(\frac{1}{k-1} + \alpha_{i1} + \beta_{i1}))]$, where the equality holds if and only if $\forall i$,

$\boldsymbol{w}_1^{\mathrm{T}} \boldsymbol{z}_i = ... = \boldsymbol{w}_{y_i-1}^{\mathrm{T}} \boldsymbol{z}_i = \boldsymbol{w}_{y_i+1}^{\mathrm{T}} \boldsymbol{z}_i = ... = \boldsymbol{w}_k^{\mathrm{T}} \boldsymbol{z}_i$, and $\boldsymbol{w}_{y_i}^{\mathrm{T}} \boldsymbol{z}_i = 1$. The condition can be simplified as $\forall i \neq j, \boldsymbol{w}_i^{\mathrm{T}} \boldsymbol{w}_j = \frac{-1}{k-1}$, and $\boldsymbol{z}_i = \boldsymbol{w}_{y_i}$ when $2 \leq d$ and $2 \leq k \leq d+1$. $\qquad\square$

## B    MORE ANALYSIS

In this section, we provide more analysis about the unified framework of margin-based losses in (3.2), Sample Margin Regularization, Largest-Margin Softmax (LM-Softmax) loss.

### B.1    A UNIFIED FRAMEWORK

A unified framework that covers A-Softmax (Liu et al., 2017) with feature normalization, NormFace (Wang et al., 2017), CosFace/AM-Softmax (Wang et al., 2018b;a) and ArcFace (Deng et al., 2019) as special cases can be formulated with hyper-parameters $m_1$, $m_2$ and $m_3$:

$$L_i' = -\log \frac{\exp(s(\cos(m_1 \theta_{iy_i} + m_2) - m_3))}{\exp(s(\cos(m_1 \theta_{iy_i} + m_2) - m_3)) + \sum_{j \neq y_i} \exp(s \cos \theta_{ij})}, \qquad \text{(B.1)}$$

where $\theta_{ij} = \angle(\boldsymbol{w}_j, \boldsymbol{z}_i)$. In the following, we provide the details of the derivation from (3.1) to (3.2)

For the parameter $m_1$, it satisfies that $\cos(m_1 \theta) \leq \cos(\theta)$ in SphereFace Liu et al. (2017). Therefore, based on the definition of the multiplicative-angular operator, we have $\cos(m_1 \theta_{iy_i} + m_2) \leq \cos(\theta_{iy_i} + m_2)$. To better understand the theoretical optimal solution, we make the constraint that $\theta_{iy_i} \in [0, \frac{\pi}{2}]$, which is reasonable because the unique minimizer of these losses, like SphereFace, CosFace, and ArcFace, should satisfy $\theta_{iy_i}^* = 0$, rather than belongs to $(\frac{\pi}{2}, \pi]$.

As for $m_2$, ArcFace did not analyze its range. Instead, we can easily derive that $0 \leq m_2 \leq \frac{\pi}{2}$. Otherwise, the minimum of ArcFace will be obtained at $\theta_{iy_i} = \pi$, since $\cos(\theta_{iy_i} + m_2) \leq \cos(\pi + m_2)$ when $m_2 > \frac{\pi}{2}$, which is ridiculous. Therefore, for $\theta_{iy_i}, m_2 \in [0, \frac{\pi}{2}]$, we have $\cos(\theta_{iy_i} + m_2) = \cos \theta_{iy_i} \cos m_2 - \sin \theta_{iy_i} \sin m_2 \leq \cos m_2 \cos \theta_{iy_i}$, which is the main derivation from (3.1) to (3.2).

### B.2    ON THE SAMPLE MARGIN REGULARIZATION AND BEYOND

The sample margin regularization term in (3.4) actually encourages the feature representation $\boldsymbol{z}$ to be similar to the corresponding prototype $\boldsymbol{w}_y$, and push $\boldsymbol{z}$ away from the most similar one of the other prototypes. This concept is similar to contrastive learning, where the most similar one of the other prototypes can be regarded as the hardest negative representation. And we also have

$$R_{\mathrm{sm}}(\boldsymbol{x}, y) \leq -\boldsymbol{w}_y^{\mathrm{T}} \boldsymbol{z} + \frac{1}{k-1} \sum_{j \neq y} \boldsymbol{w}_j^{\mathrm{T}} \boldsymbol{z}, \qquad \text{(B.2)}$$

where the right side can be regarded as pushing $\boldsymbol{z}$ away from the centroid $\frac{1}{k-1} \sum_{j \neq y} \boldsymbol{w}_j$ or pushing $\boldsymbol{z}$ away from other negative representations. Intuitively, we can also use the right side of Eq. B.2 as a sample margin regularization.

### B.3    MORE CLARIFICATIONS

As shown in the main paper, GM-Softmax loss, LM-Softmax loss, Sample Margin regularization, and Zero-centroid regularization serve different purposes. More specifically,

- The GM-Softmax loss is only derived as a theoretical formulation, which is not used for practical implementation.

- The LM-Softmax loss is tailored to obtain large margins with only one hyper-parameter. It can be used to replace popular margin-based losses, such as CosFace, and ArcFace, to obtain better discriminativeness of feature representations. Compared with NormFace Wang et al. (2017), LM-Softmax achieves much better performance on the task of person ReID, as shown in Table 3. This demonstrates that removing the term $\exp(s \boldsymbol{w}_y^{\mathrm{T}} \boldsymbol{z})$ in the denominator is helpful, which enforces LM-Softmax to have a stronger fitting ability.

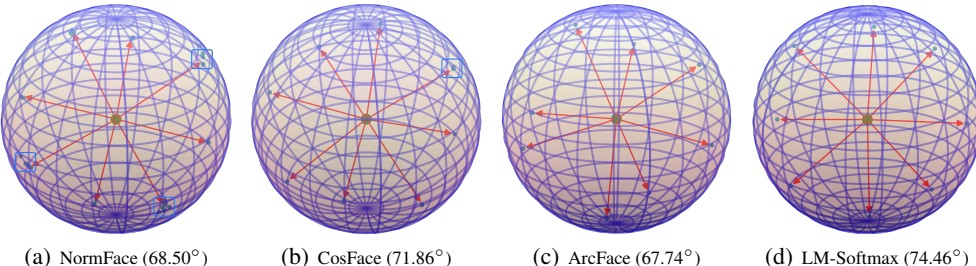

(a) NormFace (68.50°)  (b) CosFace (71.86°)  (c) ArcFace (67.74°)  (d) LM-Softmax (74.46°)

Figure 3: Visualization of the learned prototypes (red arrows) and features (green points) using NormFace, CosFace, ArcFace and LM-Softmax on $\mathbb{S}^2$ for eight classes. The optimal solution of Tammes problem for $N = 8$ have the class margin 74.86° (Whyte, 1952), where the class margin of learning with the losses NormFace, CosFace, ArcFace and LM-Softmax are 68.50°, 71.86°, 67.74° and 74.46°, respectively. We note that this phenomenon coincides with the recent popular concept—*neural collapse* (Papyan et al., 2020).

- The sample margin regularization $R_{sm}$ serves as a general regularization term to significantly improve the ability of learning towards the largest margins by combining it with the commonly-used losses. Sample margin is not new, but to the best of our knowledge, we are the first one to use it in deep learning to obtain feature representations with inter-class separability and intra-class compactness. Although theoretically learning with $R_{sm}$ can achieve the largest margins, we verify by experiments that directly maximizing sample margin cannot optimize neural networks well on complex datasets, such as CIFAR-100, as shown in Table 5. It can be found that learning with $R_{sm}$ suffers from the underfitting problem on CIFAR-100, whose performance is much worse than CE. Alternatively, we turn to use $R_{sm}$ as a regularization term, which can significantly improve the performance of commonly-used CE loss. These results demonstrate that using sample margin as the regularization term is more beneficial than using it as the loss. This is our new contribution to the classical sample margin.

- The zero-centroid regularization $R_{w}$ is specially tailored for class-imbalanced cases, which is only applied to prototypes at the last inner-product layer. Therefore, it can be easily embedded into the DNN-based methods to handle class imbalance.

## C  EXPERIMENTS

In this section, we provide the experimental details, including datasets, network architectures, parameter settings, analysis, and more results. All codes are implemented by PyTorch (Paszke et al., 2019).

We first recall the sample margin regularization $R_{sm} = -\boldsymbol{w}_y^{\mathrm{T}}\boldsymbol{z} + \max_{j \neq y} \boldsymbol{w}_j^{\mathrm{T}}\boldsymbol{z}$ and the zero-centroid regularization $R_{w} = \|\frac{1}{k}\sum_{i=1}^{k}\boldsymbol{w}_i\|_2^2$, which are used to enlarge margins for baseline methods. As for the trade-off parameter settings $\mu$ and $\lambda$ in the following experiments. We use $\mu$ and $\lambda' = 100\lambda$ denote the trade-off parameters for $R_{sm}$ and $R_{w}$, *i.e.*, $L + \mu R_{sm}$ and $L + 100\lambda R_{w}$, respectively. In the following, we set $\mu = 0.5, 1.0$ for $R_{sm}$, and $\lambda = 1, 2, 5, 10, 20$ for $R_{w}$.

### C.1  TOY EXPERIMENT

We conduct a toy experiment to show the inter-class separability and intra-class compactness using different losses, where we randomly generate prototypes $W \in \mathbb{R}^{k \times d}$ (we set $d = 3$ and $k = 8$), and initialize features $Z \in \mathbb{R}^{N \times d}$ (we set $N = 10k$). Our goal is to optimize both $W$ and $Z$ to learn the largest class margin and sample margin with different losses. According to the Tammes problem for $N = 8$, the optimal solution of $W$ and $Z$ satisfies that $m_c(W) = 74.86°$ (Whyte, 1952). The number of training epochs is set 500,000. We use cosine learning rate annealing with $T_{\max}$=10,000, and SGD optimizer with momentum 0.9 and weight decay $1e - 4$.

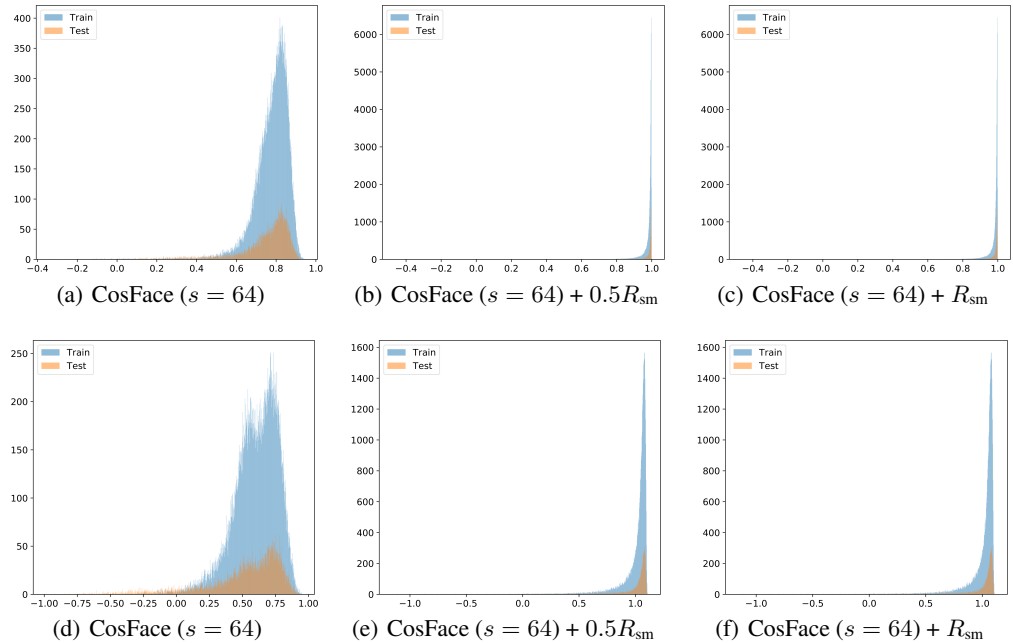

Figure 4: Histogram of similarities and sample margins for CosFace with/without sample margin regularization $R_{sm}$ on CIFAR-10. (a-c) denote the cosine similarities between samples and their corresponding prototypes, and (d-f) denote the sample margins.

**Results.** We use green points and red arrows to denote the learned feature vectors and prototype vectors, respectively. As shown in Fig. 3, the learned prototypes are separated well with Norm-Face, CosFace, ArcFace, and LM-Softmax. Specifically, the class margin of learning with the losses NormFace, CosFace, ArcFace, and LM-Softmax are 68.50°, 71.86°, 67.74° and 74.46°, respectively. As we can seen, ArcFace has a smaller class margin (67.74°) than the others, and the intra-class compactness for NormFace and CosFace is worse than LM-Softmax. The features in the blue box of Fig. 3(a) and Fig. 3(b) are not compact enough, but ArcFace and LM-Softmax do. Moreover, our proposed LM-Softmax shows better performance in class margin and sample margins, where the learned prototypes have the class margin close to the theoretical optima, and the features are perfectly optimized to be their corresponding prototypes.

## C.2 VISUAL CLASSIFICATION

We introduce three metrics to evaluate whether a loss function owns good inter-class separability and intra-class compactness. The first one is the top-1 test accuracy $acc$ to measure the generalization of the trained models. The second one is the class margin $m_{cls}$ defined in Eq. (2). And the last one we define as the average of sample margins with cosine similarities, *i.e.*, $m_{samp} = \frac{1}{N}\sum_{i=1}^{N}\frac{\boldsymbol{w}_{y_i}^T\phi_{\Theta}(\boldsymbol{x}_i)}{\|\boldsymbol{w}_{y_i}\|\|\phi_{\Theta}(\boldsymbol{x}_i)\|} - \max_{j\neq y_i}\frac{\boldsymbol{w}_{y_i}^T\phi_{\Theta}(\boldsymbol{x}_i)}{\|\boldsymbol{w}_j\|\|\phi_{\Theta}(\boldsymbol{x}_i)\|}$. Then we experiments with a 4-layer CNN, ResNet-18 and ResNet-34 (He et al., 2016) on MNIST (LeCun et al., 1998), CIFAR-10 and CIFAR-100 (Krizhevsky & Hinton, 2009), respectively. Moreover, some commonly-used neural layers are considered, such as ReLU (Glorot et al., 2011), BatchNorm (Ioffe & Szegedy, 2015), and cosine learning rate annealing (Loshchilov & Hutter, 2016).

**Datasets.** We empirically investigate the performance of learning towards the largest margins on benchmark datasets including MNIST (LeCun et al., 1998), CIFAR-10 and CIFAR-100 (Krizhevsky & Hinton, 2009).

**Training details.** We use a simple CNN which consists of *Conv(1, 32, 3) → BatchNorm (Ioffe & Szegedy, 2015) → ReLU (Glorot et al., 2011) → MaxPool(2,2) → Conv(32, 64, 3) → BatchNorm → ReLU → MaxPool(2,2) → Linear()* for MNIST, a ResNet-18 (He et al., 2016) for CIFAR-10, and a ResNet-34 (He et al., 2016) for CIFAR-100. The number of training epochs is set 100, 200 and

Table 5: Test accuracies, class margins and sample margins on MNIST, CIFAR-10 and CIFAR-100 using loss functions with/without sample margin regularization $R_{\mathrm{sm}}$, where we simply set the regularization parameter to $0.5$. The results with positive gains are **highlighted**.

| Dataset | MNIST | | | CIFAR-10 | | | CIFAR-100 | | |
|---|---|---|---|---|---|---|---|---|---|
| Metric | $acc$ | $m_{cls}$ | $m_{samp}$ | $acc$ | $m_{cls}$ | $m_{samp}$ | $acc$ | $m_{cls}$ | $m_{samp}$ |
| CE | 99.11 | 87.39° | 0.5014 | 94.12 | 81.73° | 0.6203 | 74.56 | 65.38° | 0.1612 |
| $R_{\mathrm{sm}}$ | 99.07 | 95.38° | 1.036 | 94.13 | 96.28° | 0.9791 | 62.08 | 58.58° | 0.3793 |
| CE + $0.5R_{\mathrm{sm}}$ | **99.13** | **95.41°** | **1.026** | **94.45** | **96.31°** | **0.9744** | **74.96** | **90.00°** | **0.4955** |
| CosFace ($s = 5$) | 99.11 | 95.85° | 1.020 | 94.02 | 96.33° | 0.9619 | 75.37 | 84.20° | 0.5037 |
| CosFace ($s = 10$) | 98.98 | 95.93° | 0.9839 | 94.39 | 96.00° | 0.9168 | 74.44 | 83.31° | 0.4578 |
| CosFace ($s = 20$) | 99.06 | 93.24° | 0.8376 | 94.13 | 91.22° | 0.7955 | 73.26 | 79.17° | 0.3078 |
| CosFace ($s = 40$) | 99.18 | 90.69° | 0.7650 | 93.84 | 76.09° | 0.7617 | 73.54 | 77.48° | 0.2380 |
| CosFace ($s = 64$) | 99.25 | 89.50° | 0.7581 | 93.53 | 64.14° | 0.6969 | 73.87 | 72.56° | 0.2233 |
| CosFace ($s = 5$) + $0.5R_{\mathrm{sm}}$ | 99.07 | 95.60° | **1.036** | **94.20** | 96.32° | **0.9740** | **75.52** | **90.41°** | **0.5230** |
| CosFace ($s = 10$) + $0.5R_{\mathrm{sm}}$ | **99.16** | 95.56° | **1.033** | **94.42** | **96.26°** | **0.9675** | 73.76 | **90.21°** | **0.5089** |
| CosFace ($s = 20$) + $0.5R_{\mathrm{sm}}$ | **99.24** | **95.41°** | **1.030** | **94.27** | **96.18°** | **0.9490** | **74.41** | **89.02°** | **0.4780** |
| CosFace ($s = 40$) + $0.5R_{\mathrm{sm}}$ | **99.32** | **95.41°** | **1.026** | **94.42** | **95.93°** | **0.9238** | **74.58** | **86.91°** | **0.4251** |
| CosFace ($s = 64$) + $0.5R_{\mathrm{sm}}$ | **99.27** | **95.35°** | **1.019** | **94.20** | **95.48°** | **0.9075** | **74.53** | **85.31°** | **0.3817** |
| CosFace ($s = 5$) + $R_{\mathrm{sm}}$ | **99.15** | 95.59° | **1.032** | **94.38** | **96.35°** | **0.9817** | 75.18 | **90.44°** | **0.5228** |
| CosFace ($s = 10$) + $R_{\mathrm{sm}}$ | **99.09** | 95.48° | **1.029** | **94.49** | **96.32°** | **0.9770** | 73.93 | **90.36°** | **0.5237** |
| CosFace ($s = 20$) + $R_{\mathrm{sm}}$ | **99.08** | **95.37°** | **1.028** | **94.36** | **96.24°** | **0.9640** | **73.79** | **89.63°** | **0.4958** |
| CosFace ($s = 40$) + $R_{\mathrm{sm}}$ | 99.12 | **95.38°** | **1.027** | **94.31** | **96.18°** | **0.9510** | **74.43** | **88.83°** | **0.4736** |
| CosFace ($s = 64$) + $R_{\mathrm{sm}}$ | 99.18 | **95.38°** | **1.025** | **94.60** | **96.02°** | **0.9443** | **74.05** | **87.83°** | **0.4390** |
| ArcFace ($s = 5$) | 99.05 | 95.46° | 0.9956 | 93.90 | 96.33° | 0.9473 | 75.08 | 78.28° | 0.4884 |
| ArcFace ($s = 10$) | 99.05 | 94.64° | 0.8225 | 94.50 | 91.23° | 0.8501 | 73.96 | 76.91° | 0.4313 |
| ArcFace ($s = 20$) | 99.11 | 90.84° | 0.6091 | 94.11 | 53.98° | 0.5707 | 74.74 | 60.91° | 0.3010 |
| ArcFace ($s = 40$) | 99.13 | 86.13° | 0.4606 | 93.88 | 35.68° | 0.3195 | — | — | — |
| ArcFace ($s = 64$) | 99.21 | 82.63° | 0.4038 | — | — | — | — | — | — |
| ArcFace ($s = 5$) + $0.5R_{\mathrm{sm}}$ | 99.00 | **95.59°** | **1.034** | **94.17** | 96.32° | **0.9731** | 74.72 | **90.37°** | **0.5081** |
| ArcFace ($s = 10$) + $0.5R_{\mathrm{sm}}$ | **99.14** | **95.42°** | **1.034** | 94.21 | **96.27°** | **0.9651** | **74.47** | **90.13°** | **0.5143** |
| ArcFace ($s = 20$) + $0.5R_{\mathrm{sm}}$ | **99.19** | **91.38°** | **1.030** | **94.32** | **96.15°** | **0.9571** | 74.64 | **88.73°** | **0.4804** |
| ArcFace ($s = 40$) + $0.5R_{\mathrm{sm}}$ | **99.24** | **95.34°** | **1.026** | 94.07 | **95.69°** | **0.9434** | — | — | — |
| ArcFace ($s = 64$) + $0.5R_{\mathrm{sm}}$ | 99.14 | **95.29°** | **1.019** | — | — | — | — | — |
| ArcFace ($s = 5$) + $R_{\mathrm{sm}}$ | **99.17** | 95.53° | **1.030** | **94.40** | **96.35°** | **0.9825** | 74.85 | **90.41°** | **0.5156** |
| ArcFace ($s = 10$) + $R_{\mathrm{sm}}$ | **99.09** | **95.37°** | **1.029** | 94.14 | **96.32°** | **0.9713** | 73.76 | **90.30°** | **0.5259** |
| ArcFace ($s = 20$) + $R_{\mathrm{sm}}$ | 99.11 | **95.36°** | **1.028** | **94.45** | **96.25°** | **0.9676** | 74.61 | **89.65°** | **0.5033** |
| ArcFace ($s = 40$) + $R_{\mathrm{sm}}$ | 99.02 | **95.34°** | **1.026** | **94.39** | **96.04°** | **0.9621** | — | — | — |
| ArcFace ($s = 64$) + $R_{\mathrm{sm}}$ | 99.13 | **95.30°** | **1.024** | — | — | — | — | — | — |
| NormFace ($s = 5$) | 99.03 | 95.68° | 0.9836 | 94.34 | 96.34° | 0.9452 | 75.56 | 85.37° | 0.5076 |
| NormFace ($s = 10$) | 99.06 | 94.34° | 0.7750 | 94.16 | 94.40° | 0.8004 | 74.23 | 79.10° | 0.4250 |
| NormFace ($s = 20$) | 99.09 | 89.27° | 0.5263 | 94.09 | 74.32° | 0.6001 | 73.87 | 77.47° | 0.2498 |
| NormFace ($s = 40$) | 99.06 | 85.44° | 0.3473 | 94.11 | 47.52° | 0.3825 | 73.73 | 66.67° | 0.1439 |
| NormFace ($s = 64$) | 99.00 | 82.08° | 0.2621 | 94.01 | 36.50° | 0.2633 | 73.42 | 52.37° | 0.0993 |
| NormFace ($s = 5$) + $0.5R_{\mathrm{sm}}$ | **99.15** | 95.55° | **1.035** | 94.11 | 96.32° | **0.9739** | 74.82 | **90.38°** | **0.5124** |
| NormFace ($s = 10$) + $0.5R_{\mathrm{sm}}$ | **99.16** | **95.38°** | **1.034** | **94.23** | **96.28°** | **0.9650** | **74.54** | **90.10°** | **0.5160** |
| NormFace ($s = 20$) + $0.5R_{\mathrm{sm}}$ | **99.19** | **95.37°** | **1.031** | **94.38** | **96.17°** | **0.9519** | **74.75** | **88.86°** | **0.4773** |
| NormFace ($s = 40$) + $0.5R_{\mathrm{sm}}$ | **99.14** | **95.36°** | **1.026** | **94.18** | **95.59°** | **0.9495** | **74.48** | **84.78°** | **0.4181** |
| NormFace ($s = 64$) + $0.5R_{\mathrm{sm}}$ | **99.34** | **95.29°** | **1.021** | **94.42** | **93.87°** | **0.9508** | **74.33** | **76.02°** | **0.3665** |
| NormFace ($s = 5$) + $R_{\mathrm{sm}}$ | **99.14** | 95.48° | **1.029** | **94.42** | 96.34° | **0.9798** | 74.89 | **90.45°** | **0.5134** |
| NormFace ($s = 10$) + $R_{\mathrm{sm}}$ | **99.12** | **95.37°** | **1.028** | **94.31** | **96.32°** | **0.9758** | 73.16 | **90.31°** | **0.5183** |
| NormFace ($s = 20$) + $R_{\mathrm{sm}}$ | 99.11 | **95.35°** | **1.028** | **94.16** | **96.25°** | **0.9656** | **74.23** | **89.72°** | **0.5004** |
| NormFace ($s = 40$) + $R_{\mathrm{sm}}$ | 99.11 | **95.36°** | **1.026** | 93.98 | **95.87°** | **0.9583** | **74.22** | **88.73°** | **0.4731** |
| NormFace ($s = 64$) + $R_{\mathrm{sm}}$ | **99.14** | **95.34°** | **1.025** | 94.04 | **94.35°** | **0.9570** | **74.24** | **81.57°** | **0.4386** |

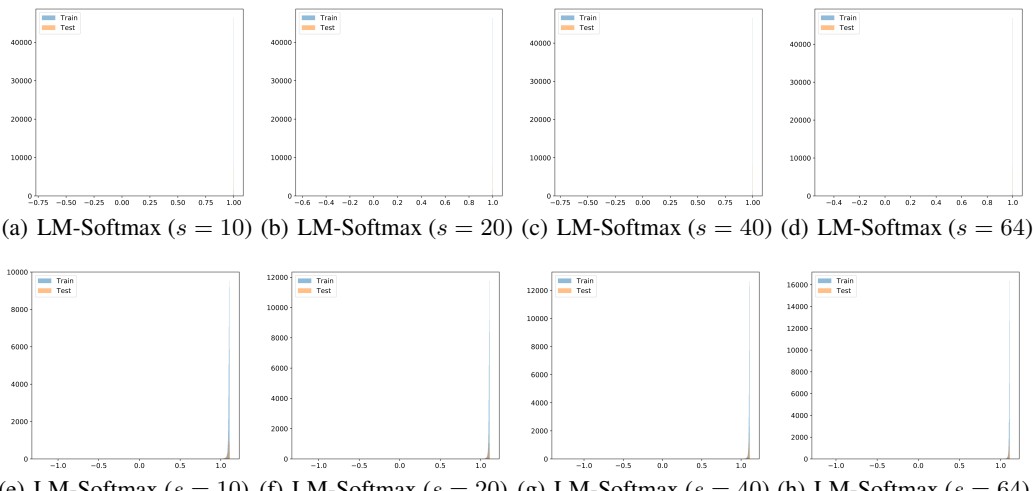

(a) LM-Softmax ($s = 10$) (b) LM-Softmax ($s = 20$) (c) LM-Softmax ($s = 40$) (d) LM-Softmax ($s = 64$)

(e) LM-Softmax ($s = 10$) (f) LM-Softmax ($s = 20$) (g) LM-Softmax ($s = 40$) (h) LM-Softmax ($s = 64$)

Figure 5: Histogram of similarities and sample margins for LM-Softmax on CIFAR-10. (a-d) denote the cosine similarities between samples and their corresponding prototypes, and (e-h) denote the sample margins.

250 for MNIST, CIFAR-10, and CIFAR-100, respectively. For all training, we use SGD optimizer with momentum 0.9 and cosine learning rate annealing (Loshchilov & Hutter, 2016) when $T_{\max}$ is equal to the corresponding epochs. Weight Decay is set to $1 \times 10^{-4}$ for MNIST, CIFAR-10, and CIFAR-100. The initial learning rate is set to 0.01 for MNIST and 0.1 for CIFAR-10 and CIFAR-100. Moreover, batch size is set to 256. Typical data augmentations including random width/height shift and horizontal flip are applied.

**Baselines and hyper-parameter settings.** We consider the baseline methods, including the commonly-used loss function CE, and margin-based loss functions NormFace, CosFace, and ArcFace with normalization for both feature vectors and class centers, and our proposed LM-Softmax loss. We have tuned their hyper-parameters for the best performance, and the specific settings are: for CosFace, we set $m = 0.1$; for ArcFace, we set $m = 0.1$. To learn towards the largest margins, we boost them with the sample margin regularization, and the trade-off parameter is set to $0.5$ and $1$. Moreover, we tune their identical hyper-parameter $s$, and show them for a comprehensive study.

**Results.** The test accuracy, class margin and the average of all sample margins are reported in Table 1. As we can see, the baseline methods fail in learning large margins for all $s$, and there is no significant difference in the performance of these losses. More specifically, the class margin decreased as $s$ increases, while the losses with the sample margin regularization $R_{\mathrm{sm}}$ usually remain the large class margins, and the class margins are close to the optimal results ($\arccos(-1/9) = 96.37°$ for MNIST and CIFAR-10, and $\arccos(-1/99) = 90.57°$ for CIFAR-100). To better describe the the inter-class separability and intra-class compactness, we provide the histograms of sample margins and similarities between the learned features and their corresponding prototype that they belong to. In Fig. 9, the similarities in Fig. 9(a) are mainly concentrated in 0.8 for CosFace with $s = 64$, while the similarities in Fig. 9(b) and 9(c) are very close to 1. This indicates that the sample margin regularization significantly improves the inter-class compactness (the learned features in the same class are very similar to their corresponding prototype.) Moreover, the histograms of our proposed LM-Softmax on CIFAR-10 and CIFAR-100 are reported in Fig. 5 and 6, respectively. The similarities and sample margins keep very large with different $s$. More visualizations are provided in the following figures.

**Clarification.** As shown in table 5, the proposed method results in both more larger class margin and more larger sample margin than the compared methods, however, the accuracy of the proposed method is slightly better than accuracies of the compared methods. $acc$ actually evaluate the proportion of samples whose sample margin is larger than 0, i.e., $acc = \frac{1}{N} \sum_{i=1}^{N} \mathbb{I}(\gamma(x_i, y_i) > 0)$. $acc$ is a good evaluation criterion for classification but is not good enough to measure the quality of feature representation. This is also one of the motivations of the previous works to improve the original

softmax loss. In this paper, we measure the inter-class separability and intra-class compactness by class margin and sample margin, which can be used as two criteria to evaluate the quality of feature representations. Thus, $acc$, class margin, and sample margin can be regarded as different criteria.

Although the relationship of $acc$ and margins is not so straightforward, enlarging the margins can improve acc to some extent. As shown in Table 1, we can see that enlarging the margins of other losses by adding the sample margin regularization $R_{sm}$ can improve the accuracy in most cases. Moreover, as shown in Table 2, the results on imbalanced learning are noteworthy, where the zero-centroid regularization for learning towards the largest margins on imbalanced classification shows obvious improvements in both class margins and accuracy in most cases, and even can improve the performance of LDAM that is tailored for imbalanced learning

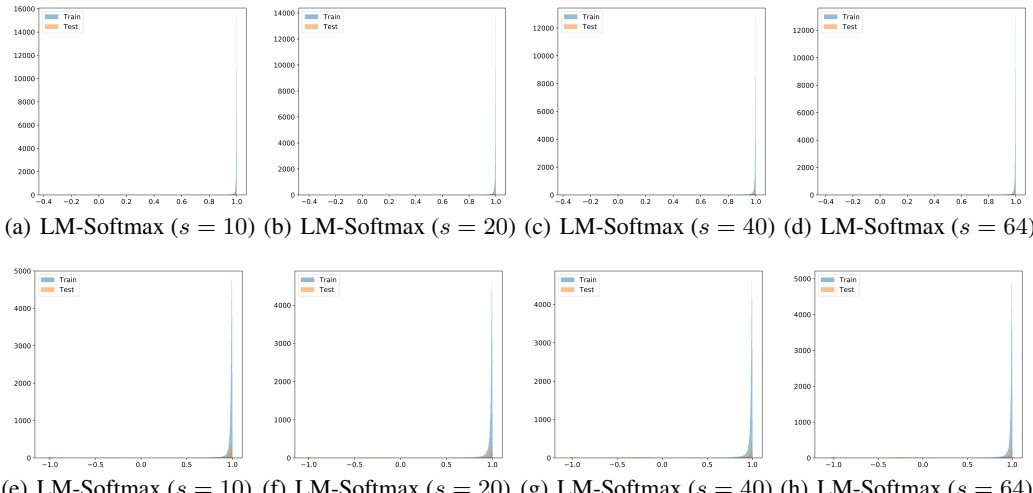

(a) LM-Softmax ($s = 10$) (b) LM-Softmax ($s = 20$) (c) LM-Softmax ($s = 40$) (d) LM-Softmax ($s = 64$)

(e) LM-Softmax ($s = 10$) (f) LM-Softmax ($s = 20$) (g) LM-Softmax ($s = 40$) (h) LM-Softmax ($s = 64$)

Figure 6: Histogram of similarities and sample margins for LM-Softmax on CIFAR-100. (a-d) denote the cosine similarities between samples and their corresponding prototypes, and (e-h) denote the sample margins.

## C.3 IMBALANCED CLASSIFICATION

**Imbalanced CIFAR-10 and CIFAR-100.** The original version of CIFAR-10 and CIFAR-100 contains 50,000 training images and 10,000 test images of size $32 \times 32$ with 10 and 100 classes, respectively. To create their imbalanced version, we follow the setting in (Buda et al., 2018; Cui et al., 2019; Cao et al., 2019), where we reduce the number of training examples per class, and keep the test set unchanged. To ensure that our methods apply to a variety of settings, we consider two types of imbalance: long-tailed imbalance (Cui et al., 2019) and step imbalance (Buda et al., 2018). We use the imbalance ratio $\rho$ to denote the ratio between sample sizes of the most frequent and least frequent class, *i.e.*, $\rho = \max_i\{n_i\} / \min_i\{n_i\}$. Long-tailed imbalance utilizes an exponential decay in sample sizes across different classes. For step imbalance setting, all minority classes have the same sample size, as do all frequent classes. This gives a clear distinction between minority classes and frequent classes, and the fraction for minority classes is defined as $\mu$. We follow (Cao et al., 2019) and set $\mu = 0.5$ by default.

We report the top-1 test accuracy $acc$ and class margin $m_{cls}$ of various baseline methods, including CE, Focal Loss, NormFace, CosFace, ArcFace, and the Label-Distribution-Aware Margin Loss (LDAM) with hyper-parameter $s = 5$. Moreover, the proposed LM-Softmax loss actually is greatly affected by data imbalance since it will pay much attention to enlarge the margin between frequent classes and minority classes than other losses rather than any two classes. And we experiment with the LM-Softmax to verify the validity of the enlarging margin method. Moreover, we add the zero-centroid regularization to the losses whose feature and prototypes are normalized for better margins.

**Training details.** We use ResNet-18 for imbalanced CIFAR-10, and ResNet-34 for imbalanced CIFAR-100. Following in (Cao et al., 2019), we use SGD optimizer with momentum 0.9 and weight decay $2 \times 10^{-4}$. The number of training epochs is set 200, and batch size is 128. The initial learning

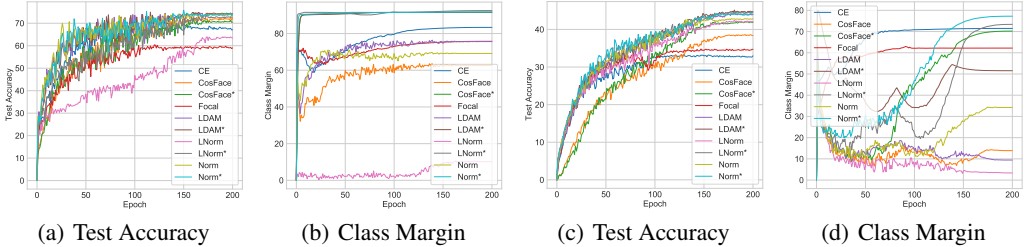

| (a) Test Accuracy | (b) Class Margin | (c) Test Accuracy | (d) Class Margin |

Figure 7: Test accuracies and class margins using different loss functions with and without the zero-centroid regularization on imbalanced CIFAR-10 and CIFAR-100. (a) and (b) are test accuracies and class margins on imbalanced CIFAR-10, respectively. (c) and (d) are test accuracies and class margins on imbalanced CIFAR-10, respectively.

rate is set to 0.1. Moreover, we use the cosine learning rate annealing strategy (Loshchilov & Hutter, 2016) when $T_{\max}$ is equal to the corresponding epochs.

**Baselines and their hyper-parameter settings.** We consider the baseline methods, including CE, Focal loss, CosFace, NormFace, ArcFace, LM-Softmax, and the label-distribution-aware margin (LDAM) loss. We set $\gamma = 1$ for Folcal, $m = 0.35$ for CosFace, $m = 0.1$ for ArcFace with stable results, and the identical hyper-parameter $s$ is set to 5.

**Results.** The experimental results of imbalanced CIFAR-10 and CIFAR-100 are reported in Table 6. As we can see, the class margin of the LM-Softmax loss is fairly low in the severely imbalanced cases, while the other losses with feature and weight normalization have better performance than CE and Focal. However, their class margins are still small. With the role of the zero-centroid regularization, the class margin has a very obvious improvement in all cases, where the class margins are close to the optimal one ($\arccos(-1/9) = 96.37°$ for imbalanced CIFAR-10, and $\arccos(-1/99) = 90.57°$ for imbalanced CIFAR-100). This conclusion holds for any choice of $\lambda$. As for the accuracy, there are also good improvements in most cases, especially for imbalanced CIFAR-100. Moreover, compared with the performance of NormFace, it is worth noticing that the improvements of LDAM may heavily rely on the features and prototype normalization even if LDAM is designed for label-distribution-aware margin trade-off. As illustrated in Fig. 20-28, the zero-centroid regularization improves the intra-class compactness, where the cosine similarities between features and their corresponding prototypes they belong to are more concentrated around 1. The experimental results on the task of imbalanced classification. In the class imbalanced scenario, the stronger fitting ability of LM-Softmax however would make the learner care more about the majority classes but neglect the minority classes. This is the reason why LM-Softmax is less stable, which can be alleviated by applying the proposed zero-centroid regularization.

**More Comparisons.** To better show the effectiveness of zero-centroid regularization, we also construct more comparison to other related works of imbalanced learning, including two-stage methods cRT (Kang et al., 2020) and MiSLAS (Zhong et al., 2021). cRT works in a two-stage manner: firstly learn feature representation from the original imbalanced data, and then retrain the classifier using class-balanced sampling with the first-stage representation frozen. Our proposed zero-centroid regularization $R_w$ can not only render zero-centroid classifier but also produce feature representations with larger margins when directly learning with imbalanced datasets. Thus, our proposed zero-centroid regularization can benefit these two-stage methods. To verify this point, we conduct experiments on ImageNet-LT with backbone ResNet-50 where the experimental settings follow a recent two-stage decoupling method MiSLAS. As shown in the following table, the performance comparison of CE and CE + $R_w$ demonstrate that zero-centroid regularization can significantly improve the representation learning ability of the first stage. Moreover, our zero-centroid regularization $R_w$ can be easily integrated into well-developed two-stage decoupling methods, such as cRT, MiS-LAS. As demonstrated by the following results, adding $R_w$ into 1st stage (representation learning only) or both stages (representation learning and classifier learning) all can improve the performance of the original methods.

Table 6: Test accuracies ($acc$) and class margins ($m_{cls}$) on imbalanced CIFAR-10. The results with positive gains are **highlighted** (where $\lambda$ denotes the regularization coefficient of the zero-centroid regularization term).

| Dataset | Imbalanced CIFAR-10 | | | | | | | | Imbalanced CIFAR-100 | | | | | | | |
|---|---|---|---|---|---|---|---|---|---|---|---|---|---|---|---|---|
| Imbalance Type | long-tailed | | | | step | | | | long-tailed | | | | step | | | |
| Imbalance Ratio | 100 | | 10 | | 100 | | 10 | | 100 | | 10 | | 100 | | 10 | |
| Metric | acc | $m_{cls}$ | acc | $m_{cls}$ | acc | $m_{cls}$ | acc | $m_{cls}$ | acc | $m_{cls}$ | acc | $m_{cls}$ | acc | $m_{cls}$ | acc | $m_{cls}$ |
| CE | 70.88 | 87.41° | 88.17 | 79.63° | 64.21 | 76.50° | 85.06 | 82.24° | 40.38 | 64.73° | 60.42 | 66.24° | 42.36 | 60.32° | 56.88 | 62.83° |
| Focal | 66.30 | 74.14° | 87.33 | 74.48° | 60.55 | 63.30° | 84.49 | 75.16° | 38.04 | 54.67° | 60.09 | 59.30° | 41.90 | 55.98° | 57.84 | 55.72° |
| CosFace ($\lambda = 0$) | 69.28 | 58.77° | 87.02 | 81.61° | 53.64 | 19.78° | 84.86 | 75.96° | 34.91 | 4.73° | 60.60 | 70.82° | 40.36 | 0.76° | 47.56 | 8.56° |
| CosFace ($\lambda = 1$) | 68.86 | **96.17°** | 87.24 | **96.16°** | 62.24 | **95.93°** | 84.98 | **96.26°** | 40.53 | **65.42°** | 60.37 | **84.84°** | 40.90 | **42.85°** | 56.50 | **74.41°** |
| CosFace ($\lambda = 2$) | **69.40** | **95.61°** | 87.16 | **96.26°** | 62.49 | **95.86°** | 84.69 | **95.88°** | 40.53 | **65.13°** | 60.77 | **84.97°** | 40.84 | **42.96°** | 56.73 | **71.08°** |
| CosFace ($\lambda = 5$) | 69.18 | **93.73°** | 87.34 | **96.24°** | 62.13 | **95.84°** | 85.07 | **96.24°** | 40.58 | **55.27°** | 60.34 | **84.47°** | 41.12 | **43.79°** | 57.22 | **75.42°** |
| CosFace ($\lambda = 10$) | 68.83 | **92.49°** | 86.94 | **96.23°** | 61.99 | **95.35°** | 85.59 | **96.12°** | 40.98 | **80.93°** | 59.15 | **85.07°** | 40.97 | **34.65°** | 56.97 | **84.09°** |
| CosFace ($\lambda = 20$) | **69.52** | **91.90°** | 87.55 | **95.46°** | 62.38 | **94.36°** | 85.15 | **95.88°** | 39.92 | **80.30°** | 59.66 | **83.46°** | 41.17 | **41.59°** | 57.97 | **83.93°** |
| ArcFace ($\lambda = 0$) | 72.20 | 65.86° | 89.00 | 85.23° | 62.48 | 54.29° | 86.32 | 80.51° | 42.77 | 13.22° | 63.21 | 67.73° | 41.47 | 0.50° | 58.89 | 0.37° |
| ArcFace ($\lambda = 1$) | 71.69 | **95.08°** | 88.86 | **96.26°** | 63.10 | **95.83°** | 86.49 | **96.23°** | 43.97 | **52.75°** | 63.67 | **71.52°** | 44.45 | **0.71°** | 61.11 | **62.38°** |
| ArcFace ($\lambda = 2$) | 71.91 | **93.78°** | 88.78 | **96.24°** | 63.05 | **94.84°** | 86.18 | **96.23°** | 44.19 | **55.95°** | 63.54 | **72.68°** | 44.41 | **0.81°** | 60.71 | **0.62°** |
| ArcFace ($\lambda = 5$) | **72.23** | **92.30°** | **89.22** | **96.23°** | 64.38 | **95.01°** | 86.56 | **96.24°** | 44.68 | **56.60°** | 63.80 | **73.45°** | 43.79 | **0.61°** | 60.30 | **63.68°** |
| ArcFace ($\lambda = 10$) | 71.99 | **91.92°** | 88.99 | **94.68°** | 63.59 | **94.97°** | 86.65 | **96.23°** | 43.89 | **75.58°** | 63.55 | **82.11°** | 44.11 | **31.54°** | 60.44 | **69.63°** |
| ArcFace ($\lambda = 20$) | 71.75 | **91.42°** | 88.99 | **92.85°** | 63.56 | **93.29°** | 86.15 | **95.83°** | 43.55 | **75.28°** | 62.10 | **81.00°** | 44.26 | **32.10°** | 60.79 | **79.85°** |
| NormFace ($\lambda = 0$) | 72.37 | 62.72° | 89.19 | 82.60° | 63.69 | 51.00° | 86.37 | 77.82° | 43.71 | 16.11° | 63.50 | 71.26° | 41.93 | 1.36° | 59.85 | 21.32° |
| NormFace ($\lambda = 1$) | 72.07 | **94.95°** | 89.18 | **96.27°** | 62.40 | **96.15°** | 86.46 | **96.29°** | 44.18 | **59.42°** | 63.81 | **79.85°** | 43.77 | **41.25°** | 61.04 | **64.55°** |
| NormFace ($\lambda = 2$) | 71.92 | **94.29°** | 88.93 | **96.28°** | 63.21 | **96.14°** | 86.26 | **96.30°** | 44.20 | **60.39°** | 63.90 | **77.69°** | 44.51 | **36.30°** | 60.49 | **71.70°** |
| NormFace ($\lambda = 5$) | 70.79 | **92.37°** | 88.84 | **96.17°** | 62.83 | **95.38°** | 86.49 | **96.28°** | 44.25 | **64.85°** | 63.60 | **77.74°** | 44.14 | **36.62°** | 60.30 | **73.08°** |
| NormFace ($\lambda = 10$) | 72.04 | **91.95°** | **89.30** | **94.50°** | 63.45 | **94.75°** | 86.06 | **96.29°** | 43.71 | **74.87°** | 63.17 | **82.71°** | 43.61 | **36.47°** | 60.22 | **80.83°** |
| NormFace ($\lambda = 20$) | 71.36 | **91.14°** | 89.08 | **93.40°** | 64.07 | **93.06°** | 86.50 | **95.94°** | 43.67 | **75.71°** | 62.66 | **82.18°** | 43.70 | **28.94°** | 60.16 | **81.66°** |
| LDAM ($\lambda = 0$) | 72.86 | 73.30° | 88.92 | 88.19° | 63.27 | 61.42° | 87.04 | 85.21° | 43.28 | 7.73° | 63.62 | 73.19° | 41.65 | 0.85° | 58.32 | 6.08° |
| LDAM ($\lambda = 1$) | 72.50 | **96.25°** | 88.97 | **96.24°** | 64.31 | **96.10°** | 86.74 | **96.26°** | 44.18 | **71.00°** | 63.95 | **84.49°** | 44.14 | **39.14°** | 60.52 | **71.43°** |
| LDAM ($\lambda = 2$) | 72.41 | **95.85°** | 89.01 | **96.24°** | 64.99 | **96.04°** | 86.55 | **96.28°** | 44.90 | **67.95°** | 64.12 | **85.81°** | 44.40 | **36.96°** | 60.83 | **75.22°** |
| LDAM ($\lambda = 5$) | 71.99 | **93.83°** | **89.51** | **96.25°** | 64.79 | **96.12°** | 86.62 | **96.16°** | 45.23 | **70.96°** | 64.18 | **85.03°** | 43.80 | **40.03°** | 60.83 | **72.27°** |
| LDAM ($\lambda = 10$) | 72.21 | **92.49°** | 88.92 | **96.18°** | 64.48 | **96.16°** | 86.69 | **96.29°** | 43.53 | **81.42°** | 63.05 | **85.62°** | 44.48 | **43.26°** | 60.39 | **83.06°** |
| LDAM ($\lambda = 20$) | 72.86 | **91.75°** | 89.20 | **95.59°** | 64.66 | **94.55°** | 86.60 | **96.05°** | 43.85 | **79.65°** | 62.64 | **84.87°** | 44.17 | **37.66°** | 60.28 | **84.31°** |
| LM-Softmax ($\lambda = 0$) | 65.32 | 4.42° | 88.69 | 68.91° | 50.47 | 0.45° | 86.08 | 52.20° | 41.52 | 4.50° | 63.26 | 68.31° | 41.53 | 0.47° | 55.44 | 1.37° |
| LM-Softmax ($\lambda = 1$) | 72.25 | **96.06°** | 88.47 | **96.26°** | 64.18 | **91.44°** | 86.66 | **96.14°** | 45.22 | **68.02°** | 63.77 | **81.99°** | 45.40 | **39.87°** | 60.57 | **73.19°** |
| LM-Softmax ($\lambda = 2$) | 72.57 | **95.83°** | 88.69 | **96.31°** | 65.58 | **93.23°** | 86.70 | **96.11°** | 44.90 | **67.90°** | 63.39 | **82.93°** | 45.17 | **38.29°** | 60.73 | **74.78°** |
| LM-Softmax ($\lambda = 5$) | 72.53 | **93.65°** | 88.60 | **96.26°** | 65.18 | **95.20°** | **87.07** | **96.05°** | 45.28 | **69.53°** | 63.60 | **83.32°** | 46.23 | **43.15°** | 60.22 | **74.37°** |
| LM-Softmax ($\lambda = 10$) | **73.21** | **92.57°** | 88.49 | **96.25°** | 65.91 | **93.84°** | 86.96 | **96.09°** | 44.13 | **78.90°** | 62.89 | **85.39°** | 45.06 | **46.69°** | 60.48 | **7.94°** |
| LM-Softmax ($\lambda = 20$) | 73.20 | **91.95°** | 89.12 | **95.73°** | 65.39 | **93.23°** | 86.95 | **96.03°** | 44.22 | **80.53°** | 63.40 | **83.80°** | 45.97 | **64.84°** | 60.23 | **76.77°** |

Table 7: Top-1 validation accuracy on ImageNet-LT, where * denotes that the results are borrowed from MiSLAS, X+$R_w$ denotes adding $R_w$ to the corresponding stages, and the trade-off parameter $\lambda$ is set 100. The results with positive gains are **highlighted**.

| Method | Many | Medium | Few | All |
|---|---|---|---|---|
| CE | 66.76 | 36.87 | 7.06 | 43.61 |
| CE+$R_w$ | **68.42** | **39.42** | **10.69** | **45.90** |
| cRT* | 62.5 | 47.4 | 29.5 | 50.3 |
| cRT+mixup* | 63.9 | 49.1 | 30.2 | 51.7 |
| cRT+mixup | 65.72 | 48.78 | 25.89 | 51.61 |
| cRT+mixup+$R_w$ (adding $R_w$ for 1st stage) | 64.03 | **49.89** | **32.81** | **52.59** |
| cRT+mixup+$R_w$ (adding $R_w$ for 1st and 2nd stage) | 64.12 | **49.99** | **32.73** | **52.65** |
| MiSLAS* | 61.7 | 51.3 | 35.8 | 52.7 |
| MiSLAS | 63.30 | 50.06 | 33.52 | 52.50 |
| MiSLAS+ $R_w$ (adding $R_w$ for 1st stage) | 63.11 | **50.56** | **34.24** | **52.76** |
| MiSLAS+$R_w$ (adding $R_w$ for 1st and 2nd stage) | 63.20 | **50.69** | **34.21** | **52.85** |

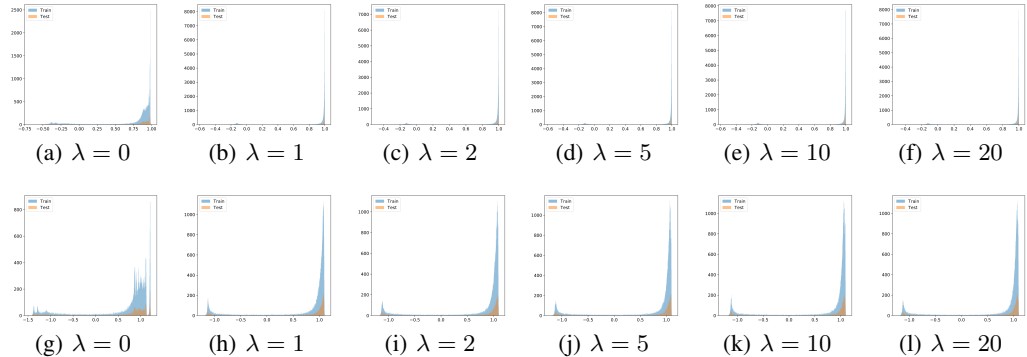

Figure 8: Histogram of similarities and sample margins for LM-Softmax using the zero-centroid regularization with different $\lambda$ on long-tailed imbalanced CIFAR-10 with $\rho = 10$. (a-f) denote the cosine similarities between samples and their corresponding prototypes, and (g-l) denote the sample margins.

## C.4 PERSON RE-IDENTIFICATION

We conduct experiments on the task of person re-identification. Specifically, we use the off-the-shelf baseline (Luo et al., 2019) as the main code to verify the efficiency of our proposed LM-Softmax.

**Training Details.** We followed the default parameter settings and training strategy. More specifically, we train the ResNet50 with pre-trained parameters for 60 epochs. Two benchmark datasets Market-1501 (Zheng et al., 2015) and DukeMTMC (Ristani et al., 2016) are evaluated. Moreover, all models are trained with Triplet Loss + the compared losses, including CE, ArcFace, CosFace, NormFace, and the proposed LM-Softmax. Experiments were conducted on Market-1501 and DukeMTMC. As shown in Table 3, our proposed LM-Softmax obtains obvious improvements in mAP, Rank@1 and Rank@5, which also exhibits significant robustness for different parameters. In contrast, ArcFace, CosFace, and NormFace show worse performance than ours and are more sensitive to parameter settings.

Table 8: The results on Market-1501 and DukeMTMC for person re-identification task. The best four results are **highlighted**.

| Dataset | Market-1501 | | | | DukeMTMC | | | |
|---|---|---|---|---|---|---|---|---|
| Method | mAP | Rank@1 | Rank@5 | Rank@10 | mAP | Rank@1 | Rank@5 | Rank@10 |
| Softmax | 82.8 | 92.7 | 97.5 | **98.7** | **73.0** | 83.5 | **93.0** | **95.2** |
| ArcFace ($s = 10$) | 67.5 | 84.1 | 92.1 | 94.9 | 37.7 | 58.7 | 72.7 | 77.8 |
| ArcFace ($s = 20$) | 79.1 | 90.8 | 96.5 | 98.1 | 61.4 | 78.3 | 88.6 | 91.6 |
| ArcFace ($s = 32$) | 80.5 | 92.1 | 97.1 | 98.4 | 66.7 | 82.9 | 91.2 | 93.4 |
| ArcFace ($s = 64$) | 80.4 | 92.6 | 97.4 | 98.4 | 67.6 | 83.4 | 91.4 | 94.1 |
| CosFace ($s = 10$) | 68.0 | 84.9 | 92.7 | 95.2 | 39.3 | 60.6 | 73.1 | 78.7 |
| CosFace ($s = 20$) | 80.5 | 92.0 | 97.1 | 98.2 | 64.2 | 81.3 | 89.7 | 92.8 |
| CosFace ($s = 32$) | 81.7 | 93.4 | **97.6** | 98.3 | 69.4 | 83.5 | 92.3 | 94.4 |
| CosFace ($s = 64$) | 78.7 | 92.0 | 97.1 | 98.3 | 68.2 | 83.1 | 92.5 | 94.4 |
| NormFace ($s = 10$) | 81.2 | 91.6 | 96.3 | 98.0 | 63.7 | 79.3 | 88.5 | 91.0 |
| NormFace ($s = 20$) | 83.2 | **93.5** | **97.9** | **98.8** | 71.6 | 83.8 | **93.3** | **95.1** |
| NormFace ($s = 32$) | 77.5 | 90.0 | 96.9 | 98.3 | 66.2 | 80.2 | 90.5 | 93.8 |
| NormFace ($s = 64$) | 77.5 | 90.0 | 96.9 | 98.3 | 60.1 | 75.2 | 88.1 | 91.7 |
| LM-Softmax ($s = 10$) | **83.3** | 92.8 | 97.1 | 98.2 | 72.2 | **85.8** | 92.4 | 94.8 |
| LM-Softmax ($s = 20$) | **84.7** | **93.8** | **97.6** | **98.6** | **74.1** | **86.4** | **93.5** | 94.9 |
| LM-Softmax ($s = 32$) | **84.3** | **93.4** | **97.7** | 98.4 | **73.3** | **86.0** | **93.2** | **95.1** |
| LM-Softmax ($s = 64$) | **84.6** | **93.9** | **98.1** | **98.8** | **74.2** | **86.6** | **93.5** | **95.2** |

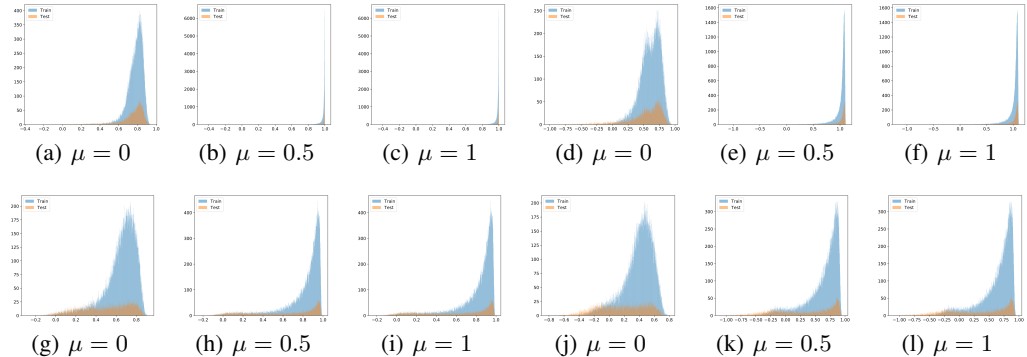

Figure 9: Histogram of similarities and sample margins for CosFace ($s = 64$) with/without sample margin regularization $R_{sm}$ on CIFAR-10 and CIFAR-100. (a-c) and (g-i) denote the cosine similarities on CIFAR-10 and CIFAR-100, respectively. (d-f) and (j-l) denote the sample margins on CIFAR-10 and CIFAR-100, respectively.

## C.5 FACE VERIFICATION

**Datasets.** We also verify our method on the task of face verification whose performance highly depends on the discriminability of feature embeddings. We follow the training settings in (An et al., 2020)[1]. The model is trained on MS1MV3 with 5.8M images and 85K ids (Guo et al., 2016) and testing on LFW (Sengupta et al., 2016), CFP-FP [3], AgeDB-30 (Moschoglou et al., 2017) and IJBC (Maze et al., 2018). The detailed results on IJBC-C are shown in Table 9

**Training Details** We use ResNet34 as the feature embedding model and train it on two GPUs NVIDIA Tesla v100 with batch size 512 for all compared methods. The compared method includes ArcFace, CosFace, NormFace, and our proposed LM-Softmax.

**Baselines and hyper-parameter settings.** We use the baseline methods including CosFace, ArcFace, NormFace, and our proposed LM-Softmax. For CosFace and ArcFace, we use the hyperparameters followed their original paper, *i.e.*, $s = 64$ and $m = 0.35$ for CosFace; $s = 64$ and $m = 0.5$ for ArcFace; For NormFace and LM-Softmax, we set $s = 64$ and $s = 32$, respectively.

Table 9: Different evaluation metrics of fave verification on IJB-C. The results with positive gains are **highlighted**.

| Method | 1e-5 | 1e-4 | AUC |
|---|---|---|---|
| ArcFace | 93.21 | 95.51 | 99.4919 |
| ArcFace+$R_{sm}$ | **93.26** | 95.41 | **99.5011** |
| ArcFace+$R_w$ | **93.27** | **95.53** | **99.5133** |
| CosFace | 93.27 | 95.63 | 99.4942 |
| CosFace+$R_{sm}$ | **93.28** | **95.68** | **99.5112** |
| CosFace+$R_w$ | **93.29** | **95.69** | **99.5538** |
| LM-Softmax | 91.85 | 94.80 | 99.4721 |
| LM-Softmax+$R_w$ | **93.17** | **95.47** | **99.5086** |

---

[1]https://github.com/deepinsight/insightface/

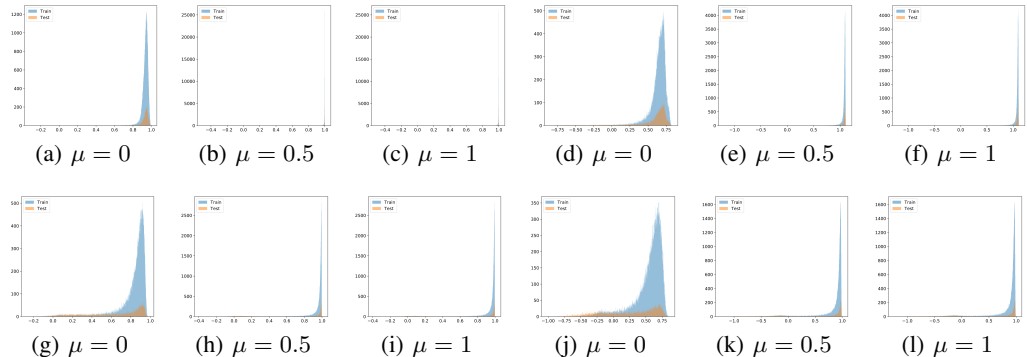

Figure 10: Histogram of similarities and sample margins for ArcFace ($s = 20$) with/without sample margin regularization $R_{\mathrm{sm}}$ on CIFAR-10 and CIFAR-100. (a-c) and (g-i) denote the cosine similarities on CIFAR-10 and CIFAR-100, respectively. (d-f) and (j-l) denote the sample margins on CIFAR-10 and CIFAR-100, respectively.

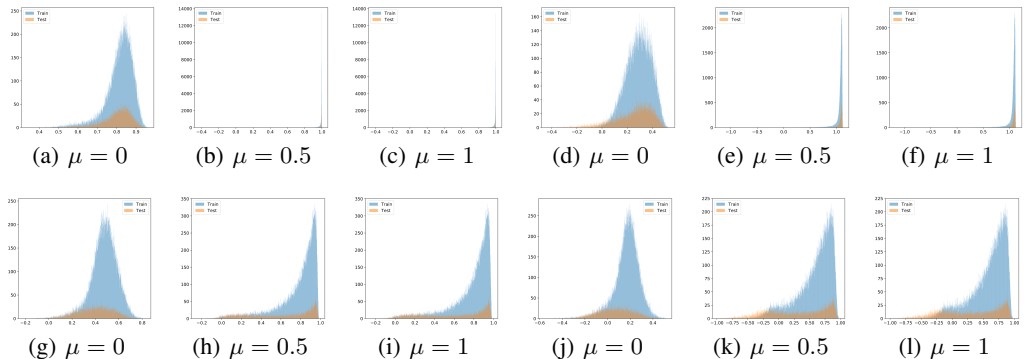

Figure 11: Histogram of similarities and sample margins for NormFace ($s = 64$) with/without sample margin regularization $R_{\mathrm{sm}}$ on CIFAR-10 and CIFAR-100. (a-c) and (g-i) denote the cosine similarities on CIFAR-10 and CIFAR-100, respectively. (d-f) and (j-l) denote the sample margins on CIFAR-10 and CIFAR-100, respectively.

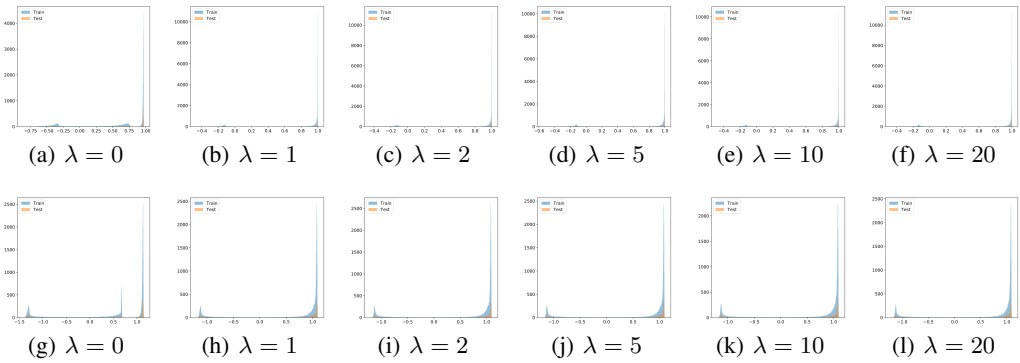

Figure 12: Histogram of similarities and sample margins for LM-Softmax using the zero-centroid regularization with different $\lambda$ on step imbalanced CIFAR-10 with $\rho = 10$. (a-f) denote the cosine similarities between samples and their corresponding prototypes, and (g-l) denote the sample margins.

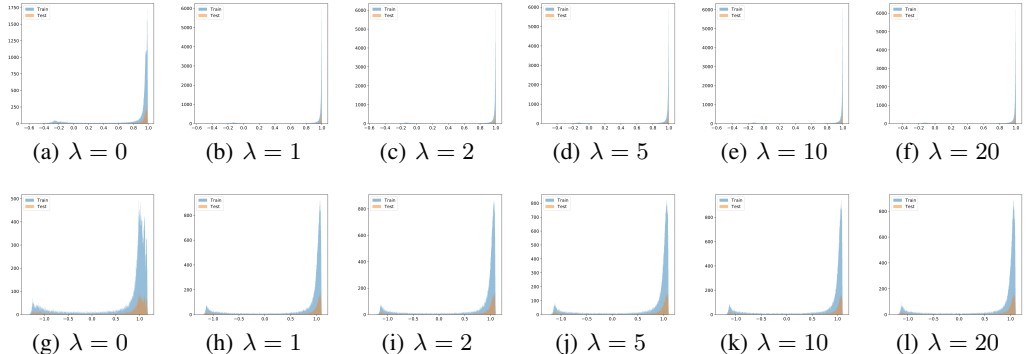

Figure 13: Histogram of similarities and sample margins for CosFace using the zero-centroid regularization with different $\lambda$ on long-tailed imbalanced CIFAR-10 with $\rho = 10$. (a-f) denote the cosine similarities between samples and their corresponding prototypes, and (g-l) denote the sample margins.

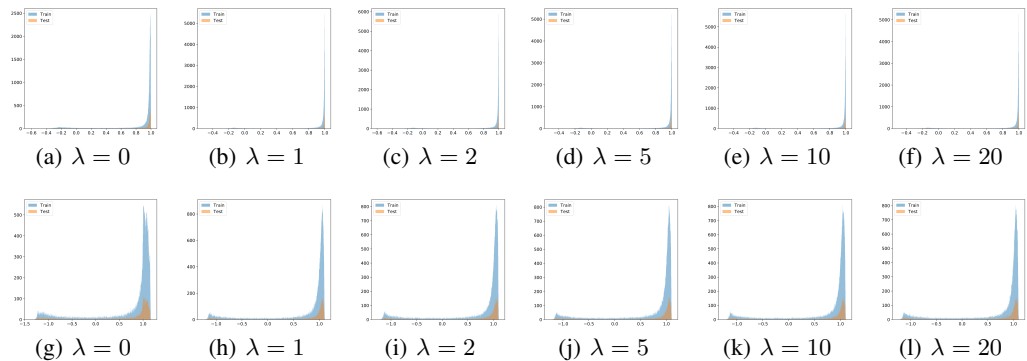

Figure 14: Histogram of similarities and sample margins for ArcFace using the zero-centroid regularization with different $\lambda$ on long-tailed imbalanced CIFAR-10 with $\rho = 10$. (a-f) denote the cosine similarities between samples and their corresponding prototypes, and (g-l) denote the sample margins.

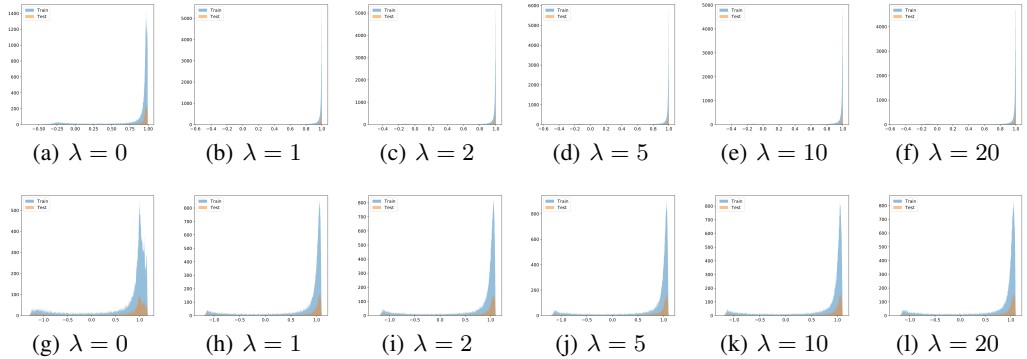

Figure 15: Histogram of similarities and sample margins for NormFace using the zero-centroid regularization with different $\lambda$ on long-tailed imbalanced CIFAR-10 with $\rho = 10$. (a-f) denote the cosine similarities between samples and their corresponding prototypes, and (g-l) denote the sample margins.

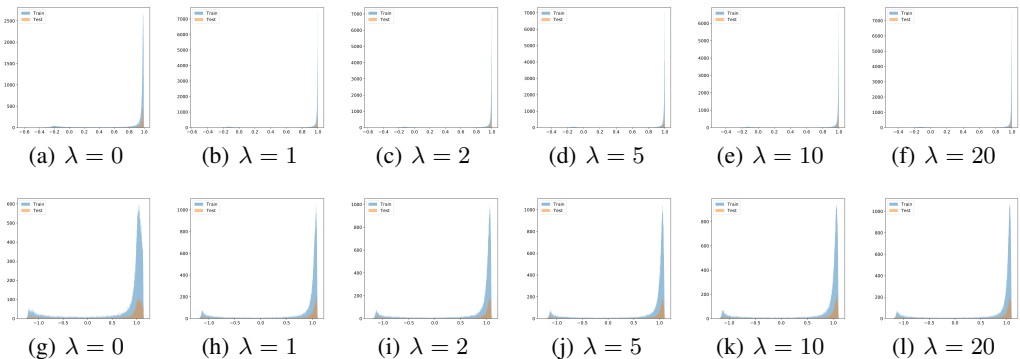

(a) $\lambda = 0$     (b) $\lambda = 1$     (c) $\lambda = 2$     (d) $\lambda = 5$     (e) $\lambda = 10$     (f) $\lambda = 20$

(g) $\lambda = 0$     (h) $\lambda = 1$     (i) $\lambda = 2$     (j) $\lambda = 5$     (k) $\lambda = 10$     (l) $\lambda = 20$

Figure 16: Histogram of similarities and sample margins for LDAM using the zero-centroid regularization with different $\lambda$ on long-tailed imbalanced CIFAR-10 with $\rho = 10$. (a-f) denote the cosine similarities between samples and their corresponding prototypes, and (g-l) denote the sample margins.

