# OpenReview forum: "Learning Towards The Largest Margins"
_ICLR.cc/2022/Conference — ICLR 2022 Poster_

### Official Review · Reviewer_3YiD · 2021-10-31

**Correctness:** 3
**Technical Novelty And Significance:** 3
**Empirical Novelty And Significance:** 3
**Recommendation:** 6
**Confidence:** 4

**Main Review:**

The strengths of the paper:
+ The paper is well-written.
+ The paper develops a principled mathematical framework and formulates it as the problem of learning towards the largest margins for better understanding and design of margin-based loss functions.
+ The sample margin regularization and zero-centroid regularization are proposed for the class-balanced case and the class-imbalanced case.
+ The paper offers rigorously theoretical analysis as well as extensive experiments to support the proposed method.

The weaknesses of the paper
- In the proposed Generalized Margin Softmax Loss (GM-Softmax), there are more parameters than a unified framework (Deng et al., 2019), how to effectively set them in the experiments?
- The improvement of the proposed is not significant. The proposed method results in both more larger class margin and more larger sample margin than the compared methods, however, the accuracy of the proposed method is slightly better than accuracies of the compared methods, and the current version does not analysis this.

**Summary Of The Paper:**

This paper develops a principled mathematical framework for better understanding and design of margin-based loss functions, where the principled optimization objective is formulated as learning towards the largest margins. In the proposed method, class margin and the sample margin are defined as the measure of inter-class separability and the measure of intra-class compactness. Furthermore, the sample margin regularization and zero-centroid regularization are introduced for the class-balanced case and the class-imbalanced case. Experimental results show the effectiveness of the proposed method on imbalanced classification, person re-identification, and face verification.

**Summary Of The Review:**

The paper is well-written and the novelty/contribution is significant and somewhat new. However, the weaknesses of the paper should be addressed to improve the paper.

---

> ### Author Response · Authors · 2021-11-14
> **Response to Reviewer 3YiD**
>
> We thank the reviewer very much for the positive comments and insightful views.
>
> **Q1**: *In the proposed Generalized Margin Softmax Loss (GM-Softmax), there are more parameters than a unified framework (Deng et al. 2019), how to effectively set them in the experiments?*
>
> **R**: Thanks for your comment. In page 6 of our paper, we stated that  "It is worth noting that we merely use the GM-Softmax loss as a theoretical formulation and will derive a more efficient form for the practical implementation." It is not necessary to consider the parameters setting of GM-Softmax, since it is just used for theoretical formulation but not for practical implementation.
>
> **Q2**: *The proposed method results in both more larger class margin and more larger sample margin than the compared methods, however, the accuracy of the proposed method is slightly better than accuracies of the compared methods, and the current version does not analysis this.*
>
> **R**: Thanks for this insightful comment. $acc$ actually evaluate the proportion of samples whose sample margin is larger than 0, i.e., $acc=\frac{1}{N}\sum_{i=1}^N\mathbb{I}(\gamma(x_i,y_i)>0)$. $acc$ is a good evaluation criterion for classification but is not good enough to measure the quality of feature representation. This is also one of the motivations of the previous works to improve the original softmax loss. In this paper, we measure the inter-class separability and intra-class compactness by class margin and sample margin, which can be used as two criteria to evaluate the quality of feature representations. Therefore, $acc$, class margin, and sample margin can be regarded as different criteria.
>
> Although the relationship of $acc$ and margins is not so straightforward, enlarging the margins can improve acc to some extent. As shown in Table 1, we can see that enlarging the margins of other losses by adding the sample margin regularization $R_{\text{sm}}$ can improve the accuracy in most cases. Moreover, as shown in Table 2, the results on imbalanced learning are noteworthy, where the zero-centroid regularization for learning towards the largest margins on imbalanced classification shows obvious improvements in both class margins and accuracy in most cases, and even can improve the performance of LDAM that is tailored for imbalanced learning

---

> > ### Comment · Reviewer_3YiD · 2021-11-30
> > **Thank you for the response**
> >
> > This paper makes a good response and clarifies the previous comments.

---

### Official Review · Reviewer_pGzf · 2021-10-31

**Correctness:** 4
**Technical Novelty And Significance:** 3
**Empirical Novelty And Significance:** 3
**Recommendation:** 8
**Confidence:** 4

**Details Of Ethics Concerns:**

There is no ethics concern about this paper.

**Main Review:**

Most of the former margin-based softmax methods (e.g. sphereface, cosface and arcface) are experiment-driven. By contrast, this paper offers rigorous theoretical analysis, which is the main contribution of this paper.

The writing of this paper is clear and all Theorems in the main paper are proved in the Supplementary Materials.

Limitations:
(1) The class margin (Eq 2.4), sample margin (Eq 3.4), generalized margin softmax loss (Eq 3.3), and zero centroid regularization (Eq 3.7) are not novel, but the theoretical explanation is nice.

(2) For face verification, which evaluation metric is employed to report the performance on IJB-C? We usually report TAR@FAR=1e-4/1e-5.

---------------post rebuttal----------------------

After reading the reply from the author, I have understood the evaluation metric used on IJB-C. I suggest the author change the AUC to TAR@FAR=1e-4/1e-5 in the camera-ready version.  The improvement is not obvious ($<0.5\\%$ IJB-C TAR@FAR=1e-4/1e-5) but it is ok for a theoretical paper. This paper can enhance our understanding of margin-based face recognition. Therefore, I still vote for acceptance.

**Summary Of The Paper:**

This paper developed a principled mathematical framework for a better understanding and design of loss functions.

Based on the class and sample margins, the proposed method formulated the objective as learning towards the largest margins, and offer rigorously theoretical analysis as support.

For class balanced cases, this paper proposed an explicit sample margin regularization term and a novel largest margin softmax loss; for the class imbalanced cases, this paper proposed a simple but effective zero centroid regularization term, which restricts the centroid of prototypes to be zero.

Extensive experimental results demonstrate that the proposed strategy significantly improves the performance in accuracy and margins for various tasks.

**Summary Of The Review:**

The theoretical explanation in this paper is nice and the experiments on person Re-Id and face verification confirm the effectiveness of the proposed method. In the NeurIPS 2021 review, this paper accepted two marginally above and one borderline result. In this ICLR submission, the main concern from the former reviewers has been fixed. In ICCV 2021, face-related papers have the highest rejection rate (>80%). There are two main groups of researchers in the face recognition community. One group of researchers focus on high performance under large-scale settings. Their papers look like experiment reports, but the performance is impressive. Another group of researchers focus on theoretical analysis. Their papers look more rigorous, but the performance is not satisfying and the proposed methods are not orthogonal to current state-of-the-art solutions. Even though I am from the first group, and I am not persuaded that the proposed method can obviously improve verification accuracy (in Table 4), I believe the theoretical explanation in this paper can enrich the research community and enhance the deep understanding of margin-based softmax loss designs. Both of the researchers from the practical and theoretical sides need to appreciate the contributions and novelties from the other side.

---

> ### Author Response · Authors · 2021-11-14
> **Response to Reviewer pGzf**
>
> We thank the reviewer for the positive and detailed review as well as the suggestions for improvement. Our response to the reviewer’s question about experiments is below:
>
> **Q**: *For face verification, which evaluation metric is employed to report the performance on IJB-C? We usually report TAR@FAR=1e-4/1e-5.*
>
> **R**: Thanks for your kind comment and suggestion. The evaluation metric used is AUC. As suggested, we report the results in terms of TAR@FAR=1e-4/1e-5. Please refer to the following results for more details, where the results with positive gains are highlighted.
>
>
> |        Method        | IJB-C (1e-5) | IJB-C(1e-4) |     AUC     |
> | :------------------: | :----------: | :---------: | :---------: |
> |       ArcFace        |    93.21     |    95.51    |   99.4919   |
> |  ArcFace ($R_{sm}$)  |  **93.26**   |    95.41    | **99.5011** |
> |  ArcFace ($R_{w}$)   |  **93.27**   |  **95.53**  | **99.5133** |
> |       CosFace        |    93.27     |    95.63    |   99.4942   |
> |  CosFace ($R_{sm}$)  |  **93.28**   |  **95.68**  | **99.5112** |
> |  CosFace ($R_{w}$)   |  **93.29**   |  **95.69**  | **99.5538** |
> |      LM-Softmax      |    91.85     |    94.80    |   99.4721   |
> | LM-Softmax ($R_{w}$) |  **93.17**   |  **95.47**  | **99.5086** |
>
> **Q**:*There are two main groups of researchers in the face recognition community. One group of researchers focus on high performance under large-scale settings. Their papers look like experiment reports, but the performance is impressive. Another group of researchers focus on theoretical analysis. Their papers look more rigorous, but the performance is not satisfying and the proposed methods are not orthogonal to current state-of-the-art solutions. Even though I am from the first group, and I am not persuaded that the proposed method can obviously improve verification accuracy (in Table 4), I believe the theoretical explanation in this paper can enrich the research community and enhance the deep understanding of margin-based softmax loss designs. Both of the researchers from the practical and theoretical sides need to appreciate the contributions and novelties from the other side.*
>
> **R**: Thanks very much for your support! We strongly agree with you. We believe high-performance algorithm and theoretical understanding are equally important to the developement of the field. Theoretical analysis can make us understand deeply about why an algorithm works well, which is helpful in inspiring further improvements.

---

### Official Review · Reviewer_pU1u · 2021-11-01

**Correctness:** 3
**Technical Novelty And Significance:** 3
**Empirical Novelty And Significance:** 2
**Recommendation:** 6
**Confidence:** 4

**Main Review:**

### PROS:
+ The authors proved that existing margin-based losses have the same lower bound and accordingly the same optimizers through connecting them to the optimizers of class-/sample-margins.
+ Some practical methods regarding classifier margins are introduced and empirically evaluated to produce favorable performance improvement on image classification tasks.

### CONS:
- Practical method.

The paper presents some practical methods. They, however, are less novel and their effectiveness are not sufficiently validated in the experiments.
Sample margin regularization in Eq.(3.4) is the classical loss such as used in MC-SVM [R1].
LM-softmax loss in Eq.(3.5) is a simple reformulation of the standard softmax loss; we can easily see that L_{softmax loss} = softplus(sL). While the softmax loss is lower-bounded by softplus, the LM-softmax loss is not by softplus but by normalization of features and classifier weights, due to which we can guess LM-softmax is less stable as shown in Table 2.
Zero-center regularization is somewhat interestingly formulated for imbalanced learning. It, however, lacks comparison/analysis to the other related works of imbalanced learning such as [R2,R3]. Besides, the regularization would not work in the framework of cRT [R4] which re-trains classifiers on 'balanced' dataset; balanced mini-batch sampling would naturally render zero-centered classifiers even by the softmax loss.

In the experiments, the authors apply simple softmax loss for comparison. However, toward fair comparison, the softmax loss should be applied to normalized features and classifiers with the tuned temperature as in the other losses. Without comparison to the well-calibrated softmax loss, it is hard to recognize the effectiveness of the proposed method, especially LM-softmax.


[R1] K. Crammer & Y. Singer. On the Algorithmic Implementation of Multiclass Kernel-based Vector Machines. JMLR, No. 2, pp.265-292, 2001.

[R2] H.-J. Ye et al. Identifying and Compensating for Feature Deviation in Imbalanced Deep Learning, arXiv:2001.01385.

[R3] B. Kim & J. Kim. Adjusting Decision Boundary for Class Imbalanced Learning, IEEE Access 2020.

[R4] B. Kang et al. Decoupling representation and classifier for long-tailed recognition. ICLR 2020.

- Theoretical analysis.

Considering the limited novelty of the presented methods as mentioned above, the paper's contribution would be rather theoretical materials. Though, those materials seem not to be so effective for further understanding/analyzing the large-margin losses.
As to normalization of features and classifiers, empirical analysis in Fig.1 is well-known in the literature of imbalanced learning [R2-R4] and Theorem 3.1 is rather trivial.
The analysis about the margin-based losses through Theorem 3.2 (Proposition 3.3) is somewhat interesting, but Theorem 3.2 is too rough to understand the roles of those losses deeply; actually, Theorem 3.2 is applicable even to the standard softmax loss with \alpha=1 and \beta=0, not only to the margin-aware ones. Thus, the theoretical part could not provide us with deep analysis specific to the margin-based losses such as XFace losses.

The paper lacks analysis and/or discussion about the case that some classes are correlated as frequently observed in real-world tasks. In that case, maximizing class-margin by force toward (k-1)-simplex classifier would be less effective, degrading intrinsic characteristics of those class categories; the correlated classes would intrinsically provide smaller margin.

### MINOR COMMENTS:
- In p.3, "intra-class compactness and inter-class separability" -> "inter-class separability and intra-class compactness"
- In Eq.3.5, -\frac{1}{s} -> \frac{1}{s}, and y_i -> y
- In Eq.3.6, \frac{1}{t} -> \frac{1}{s}
- In Table 1, no comparison to CE+R_{sm} nor LM-softmax.
- In Sec.4.3, it is unclear why the authors consider the task of person reID. How much effective is it for evaluating the margin-based losses?
- In p.13, Lemma 2.3 -> Lemma 2.1
- At the equations in top of p.15, "k \log" -> "\log".

**Summary Of The Paper:**

This paper analyzes large-margin based loss functions. The authors formulate two types of margins, class- and sample-margins, and then analyze lower bounds of various margin-based losses in a unified framework to show that those losses are minimized by the optimizers of class-/sample-margins. While inspecting the existing margin-based losses, they also formulate two practical methods of sample-margin regularization and largest margin softmax loss to further enhance margins of classifiers. The experimental results on some image classification tasks demonstrate that those methods contribute to performance improvement, while favorably being compared with the other methods that induce large-margin classification.

**Summary Of The Review:**

This paper presents several theoretical and practical materials regarding large-margin based losses.
They, however, are of limited novelty and are less contributive to theoretical understanding of margin-based losses.
Thus, my rating score of the paper is leaning toward weak rejection.


-- After rebuttal --

I appreciate the authors' effort to provide detailed response.
It nicely addresses my concerns about Theorem 3.2 and the practical methods.
Based on the rebuttal, I upgrade my score to 'weak accept' and recommend the authors to put those materials into a revised paper for clarifying the theoretical and practical contributions.

---

> ### Author Response · Authors · 2021-11-14
> **Response to Reviewer pU1u (1/3)**
>
> We thank the reviewer for the valuable comments as well as the suggestions for improvement. Our responses to the reviewer’s comments are below:
>
> **Q1**: *Sample margin regularization in Eq. (3.4) is the classical loss such as used in MC-SVM [R1].*
>
> **R**: Thanks for your comment. Sample margin is not new, but to the best of our knowledge, we are the first one to use it in deep learning to obtain feature representations with inter-class separability and intra-class compactness. Although theoretically learning with $R_{\text{sm}}$ can achieve the largest margins, we verify by experiments that directly maximizing sample margin cannot optimize neural networks well on complex datasets, such as CIFAR-100, as shown in the following table (test accuracy/class margin/mean sample margin). It can be found that learning with $R_{\text{sm}}$ suffers from the underfitting problem on CIFAR-100, whose performance is much worse than CE. Alternatively, we turn to use $R_{\text{sm}}$ as a regularization term, which can significantly improve the performance of commonly-used CE loss. These results demonstrate that using sample margin as the regularization term is more beneficial than using it as the loss. This is our new contribution to the classical sample margin.
>
> |       Method       |       MNIST        |      CIFAR-10      |     CIFAR-100      |
> | :----------------: | :----------------: | :----------------: | :----------------: |
> |         CE         | 99.11/87.39/0.5014 | 94.12/81.73/0.6203 | 74.56/65.38/0.1612 |
> |  $R_{\text{sm}}$   | 99.07/95.38/1.036  | 94.13/96.28/0.9791 | 62.08/58.58/0.3793 |
> | CE+$R_{\text{sm}}$ | 99.13/95.41/1.026  | 94.45/96.31/0.9744 | 74.96/90.00/0.4955 |
>
> **Q2**: *LM-softmax loss in Eq.(3.5) is a simple reformulation of the standard softmax loss; we can easily see that L_{softmax loss} = softplus(sL). While the softmax loss is lower-bounded by softplus, the LM-softmax loss is not by softplus but by normalization of features and classifier weights, due to which we can guess LM-softmax is less stable as shown in Table 2.*
>
> **R**:  Thanks for your comment. In our paper, we have stated that "With respect to the original softmax loss, LM-Softmax removes the term $exp(s\boldsymbol{w}_y^T\boldsymbol{z})$ in the denominator." Compared with the standard softmax loss, the proposed LM-Softmax makes two changes: 1)  performing normalization on both features and classifier weights; 2) removing the denominator. Both of them offer benefits in learning representations with inter-class separability and intra-class compactness.
>
> Firstly, in Theorem 3.2, we demonstrate in theory why the normalization of features and classifier weights is useful in learning with margin-based losses.  Before us, no works provide such theoretical understanding. Secondly, compared with NormFace [1], which is a variant of standard softmax loss with normalization on both features and classifier weights, our proposed LM-Softmax achieves much better performance on the task of person ReID, as shown in Table 3 in our paper. This demonstrates that removing the denominator of the softmax loss is also helpful, which enforces LM-Softmax to have a stronger fitting ability.
>
> Table 2 shows the experimental results on the task of imbalanced classification. In the class imbalanced scenario, the stronger fitting ability of LM-Softmax however would make the learner care more about the majority classes but neglect the minority classes. This is the reason why LM-Softmax is less stable as shown in Table 2. This issue can be alleviated by applying the proposed zero-centroid regularization, as also shown in Table 2.

---

> ### Author Response · Authors · 2021-11-14
> **Response to Reviewer pU1u (2/3)**
>
> **Q3**: *Zero-center regularization is somewhat interestingly formulated for imbalanced learning. It, however, lacks comparison/analysis to other related works of imbalanced learning such as [R2, R3]. Besides, the regularization would not work in the framework of cRT [r4] which re-trains classifier on 'balanced' dataset; balanced mini-batch sampling would naturally render zero-centered classifiers even by the softmax loss*.
>
> **R**: Thanks for your kind suggestion. In Table 2, we have compared our method with LDAM, which is a well-known margin-based loss specially tailored for imbalanced learning. The results in Table 2 can be used to verify the performance comparison of our proposed zero-centroid regularization $R_{\text{w}}$ with other related work of imbalanced learning.
>
> cRT works in a two-stage manner: firstly learn feature representation from the original imbalanced data, and then retrain the classifier using class-balanced sampling with the first-stage representation frozen. Our proposed zero-centroid regularization $R_{\text{w}}$ can not only render zero-centroid classifier but also produce feature representations with larger margins when directly learning with imbalanced datasets. Thus, our proposed zero-centroid regularization can benefit these two-stage methods. To verify this point, we conduct experiments on ImageNet-LT with backbone ResNet-50 where the experimental settings follow a recent two-stage decoupling method MiSLAS [2]. As shown in the following table, the performance comparison of CE and CE + $R_{\text{w}}$ demonstrate that zero-centroid regularization can significantly improve the representation learning ability of the first stage. Moreover, our zero-centroid regularization $R_{\text{w}}$ can be easily integrated into well-developed two-stage decoupling methods, such as cRT, MiSLAS. As demonstrated by the following results, adding $R_w$ into 1st stage (representation learning only) or both stages (representation learning and classifier learning) all can improve the performance of the original methods.
>
> |                            Method                            |   Many    |  Medium   |    Few    |    All    |
> | :----------------------------------------------------------: | :-------: | :-------: | :-------: | :-------: |
> |                              CE                              |   66.76   |   36.87   |   7.06    |   43.61   |
> |                     CE + $R_{\text{w}}$                      | **68.42** | **39.42** | **10.69** | **45.9**  |
> |                             cRT*                             |   62.5    |   47.4    |   29.5    |   50.3    |
> |                          cRT+mixup*                          |   63.9    |   49.1    |   30.2    |   51.7    |
> |                          cRT+mixup                           |   65.72   |   48.78   |   25.89   |   51.61   |
> |    cRT+mixup+$R_{\text{w}}$ ( adding $R_w$ for 1st stage)    |   64.03   | **49.89** | **32.81** | **52.59** |
> | cRT+mixup+$R_{\text{w}}$ ( adding $R_w$ for 1st and 2nd stage) |   64.12   | **49.99** | **32.73** | **52.65** |
> |                           MiSLAS*                            |   61.7    | **51.3**  | **35.8**  |   52.7    |
> |                            MiSLAS                            |   63.30   |   50.06   |   33.52   |   52.50   |
> |     MiSLAS+$R_{\text{w}}$ ( adding $R_w$ for 1st stage)      |   63.11   | **50.56** | **34.24** | **52.76** |
> | MiSLAS+$R_{\text{w}}$ ( adding $R_w$ for 1st and 2nd stage)  |   63.20   | **50.69** | **34.21** | **52.85** |
>
> where * denotes that the results are borrowed from MiSLAS [2], X+$R_{\text{w}}$ denotes adding $R_{\text{w}}$ to the corresponding stages, and the trade-off parameter $\lambda$ is set 100.
>
> **Q4**: *In the experiments, the author apply simply softmax loss for comparison. However, toward fair comparison, the softmax loss should be applied to normalized features and classifier with the tuned temperature as in the other losses.*
>
> **R**:  Thanks for your comment. NormFace [1] is what you suggest, which performs normalization on both features and classifier weights as well as temperature parameter tuning. Please refer to Table 1-4 in our paper for more details.
>
> **Q5**: *As to normalization of features and classifiers, empirical analysis in Fig.1 is well-known in the literature of imbalanced learning [R2-R4] and Theorem 3.1 is rather trivial.*
>
> **R**:  Thanks for your comment. Fig.1 is just used to explain why in the definition of class margin we do not take the magnitude into account, which is not the main contribution of this work. Theorem 3.1 is also not our main contribution, which is just used for smooth logic in paper writing. Theorem 3.1 explains the necessity of the normalization of both features and classifier weights, according to which the existing margin-based losses are concluded.

---

> ### Author Response · Authors · 2021-11-14
> **Response to Reviewer pU1u (3/3)**
>
>
> **Q6**: *The analysis about the margin-based losses through Theorem 3.2 (Proposition 3.3) is somewhat interesting, but Theorem 3.2 is too rough to understand the roles of those losses deeply; actually, Theorem 3.2 is applicable even to the standard softmax loss with \alpha=1 and \beta=0, not only to the margin-aware ones. Thus, the theoretical part could not provide us with deep analysis specific to the margin-based losses such as XFace losses.*
>
> **R**:  Thanks for your comment. We feel a little confused about this comment. There are two cases about your question:
>
> 1) "the standard softmax loss with \alpha=1 and \beta=0" is the one without normalization. In this case, "the standard softmax loss with \alpha=1 and \beta=0" is not included by Theorem 3.2, in the conditions of which we clearly state features and classifier weights should belong to the unit sphere $\mathbb{S}^{d-1}$.
>
> 2) "the standard softmax loss with \alpha=1 and \beta=0" is the one with normalization. In this case, "the standard softmax loss with \alpha=1 and \beta=0" is actually the NormFace [1], which is a representative work of margin-based losses. Proposition 3.3 actually reveals that the margin-based losses, such as NormFace, CosFace, and ArcFace, share the same optimal solution, i.e., they coincide with the goal of learning towards the largest margins.
>
> **Q7**: *The paper lacks analysis and/or discussion about the case that some classes are correlated as frequently observed in real-world tasks. In that case, maximizing class-margin by force toward (k-1)-simplex classifier would be less effective, degrading intrinsic characteristics of those class categories; the correlated classes would intrinsically provide smaller margin.*
>
> **R**: Thanks for your valuable suggestion. It is interesting and important to investigate what the margin distributions look like when considering correlated categories. We will pursue this issue in future research.
>
> It is worth noting that, without any prior knowledge about categories correlation, it is hard to automatically learn the correlation among categories for most losses (maybe contrastive learning can do it). To be specific to margin-based losses, according to Theorem 3.2 (and proposition 3.3), learning with margin-based losses can only lead to the largest margins but cannot find the correlation among categories.
>
> [1] Wang F, Xiang X, Cheng J, et al. Normface: L2 hypersphere embedding for face verification[C]//Proceedings of the 25th ACM international conference on Multimedia. 2017: 1041-1049.
>
> [2] Zhong Z, Cui J, Liu S, et al. Improving Calibration for Long-Tailed Recognition[C]//Proceedings of the IEEE/CVF Conference on Computer Vision and Pattern Recognition. 2021: 16489-16498.

---

### Official Review · Reviewer_sx1i · 2021-11-01

**Correctness:** 4
**Technical Novelty And Significance:** 3
**Empirical Novelty And Significance:** 3
**Recommendation:** 8
**Confidence:** 4

**Main Review:**

Strengths:

1. The paper is well-written, with nice motivations and clear theorems (no unnecessary "mathification" of proofs or writing).

2. Analytical results are clear and to-the-point, showing the motivation for their losses.

3. Empirical results are extensive, and show the benefits of their margin-promoting losses.

Weaknesses:

1. It would have been nice to show at least some empirical result for the case when k > d+1 (large number of classes compared to embedding dimensionality). Do their losses work "reasonably" in this case? Such a scenario may arise in modern deep learning methods when we train on datasets with many fine-grained classes e.g. ImageNet-21k.

2. It's slightly awkward that different losses are used for different datasets. How can we decide whether to use GM-Softmax, LM-Softmax or R_sm for a given dataset? Why not try all 3 together?


**Summary Of The Paper:**

This paper analyzes novel loss functions to promote large margins during the training of a supervised (softmax-based) classification loss, under various conditions such as balanced and imbalanced number of samples per class. They propose a number of novel losses to promote large margins, and show analytical and empirical proof of the optimality of their proposed losses.

**Summary Of The Review:**

The paper as it's well written, and shows novel analytical and empirical results. I think the losses are simple and easy to implement in practice.

---

> ### Author Response · Authors · 2021-11-14
> **Response to Reviewer sx1i**
>
> We thank the reviewer for the positive review and insightful questions. Answers to specific points are below:
>
> **Q1**: *It would have been nice to show at least some empirical result for the case when k > d+1 (large number of classes compared to embedding dimensionality). Do their losses work "reasonably" in this case? Such a scenario may arise in modern deep learning methods when we train on datasets with many fine-grained classes e.g. ImageNet-21k.*
>
> **R**: Thanks for the kind suggestion. In the experiments of our paper, we provide empirical results on the task of face verification. The training dataset used is MSIMV3, which includes 85K identities. Obviously, the number of classes is much larger than the embedding dimension 512 (ResNet-34), i.e., k > d+1. As shown in Table 4,  our method still works well.  We hope this experiment could answer your concern about this case.
>
> Moreover, to show the inter-class separability and intra-class compactness using different losses when $k>d+1$, we also provide a toy experiment ($k=8,d=3$) in the appendix. As illustrated in Fig. 3 of the appendix, the learned prototypes and feature representations by the proposed LM-Softmax are separated well. Among all compared losses, LM-Softmax achieves the largest margins $74.46^\circ$, which is very close to the theoretical optimal value $74.86^\circ$.
>
> **Q2**: *It's slightly awkward that different losses are used for different datasets. How can we decide whether to use GM-Softmax, LM-Softmax or R_sm for a given dataset? Why not try all 3 together?*
>
> **R**: Thanks for the kind suggestion. We would like to clarify the design purposes of these losses and regularizations, which offer the principle to decide when to use them.
>
> 1) The GM-Softmax loss is only derived as a theoretical formulation, which is not used for practical implementation.
> 2) The LM-Softmax loss is tailored to obtain large margins with only one hyper-parameter $s$. It can be used to replace popular margin-based losses, such as CosFace, and ArcFace, to obtain better discriminativeness of feature representations.
> 3) The sample margin regularization $R_{\text{sm}}$ Serves as a general regularization term to significantly improve the ability of learning towards the largest margins by combining it with the commonly-used losses. $R_{\text{sm}}$ is suggested to be used in class-balanced cases.
> 4) The zero-centroid regularization $R_{\text{w}}$ is specially tailored for class-imbalanced cases, which is only applied to prototypes at the last inner-product layer. Therefore, it can be easily embedded into the DNN-based methods to handle class imbalance.

---

### Decision · Program_Chairs · 2022-01-20

**Decision:**

Accept (Poster)

**Comment:**

This work presents a principled objective function for large margin learning. Specifically, it introduces class margin and sample margin, both of which it aims to promote. It also derives a generalized margin softmax loss which to draw general conclusions on the existing margin-based losses. The effectiveness of the proposed theory is empirically verified in visual classification, imbalanced classification, person re-identification, and face verification.

The reviewers initially raised some concerns, but most of them were well addressed in the rebuttal and convinced the reviewers. Specifically, pU1u was satisfied by authors' reply on Theorem 3.2 and the practical methods. pGzf appreciated clarifications around the evaluation metric used on IJB-C and believes this work can improve our understanding of margin-based face recognition. Finally, 3YiD had some reservations about number of parameters which got clarified by the authors.

In sum, all post rebuttal ratings fall in the accept zone, and the reviewers find the paper interesting and insightful. In concordance with them, I recommend this paper for publication. Please make sure to include suggestions made by reviewers in the camera ready version.